# 2000 years of annual ice core data from Law Dome, East Antarctica

Lenneke M. Jong[1,2], Christopher T. Plummer[2], Jason L. Roberts[1,2], Andrew D. Moy[1,2], Mark A. J. Curran[1,2], Tessa R. Vance[2], Joel K. Pedro[1,2], Chelsea A. Long[2], Meredith Nation[1,2], Paul A. Mayewski[3], and Tas D. van Ommen[1,2]

[1]Australian Antarctic Division, Department of Climate Change, Energy, the Environment and Water, Kingston, Tasmania, Australia
[2]Australian Antarctic Program Partnership, Institute of Marine and Antarctic Studies, University of Tasmania, Hobart, Tasmania, Australia
[3]Climate Change Institute, University of Maine, Orono, ME, USA

**Correspondence:** Lenneke Jong (lenneke.jong@aad.gov.au)

**Abstract.** Ice core records from Law Dome in East Antarctica collected over the last four decades, provide high resolution data for studies of the climate of Antarctica, Australia and the Southern and Indo-Pacific Oceans. Here we present a set of annually dated records of trace chemistry, stable water isotopes and snow accumulation from Law Dome covering the period from -11 to 2017 CE (1961 to -66 BP 1950), as well as the level 1 chemistry data from which the annual chemistry records are derived. Law Dome ice core records have been used extensively in studies of the past climate of the Southern Hemisphere, as well as in large scale data syntheses and reconstructions in a region where few records exist, especially at high temporal resolution. This dataset provides an update and extensions both forward and back in time of previously published subsets of the data, bringing them together into a coherent set with improved dating to enable continued use of this record. The data are available for download from the Australian Antarctic Data Centre at https://doi.org/10.26179/5zm0-v192 (Curran et al., 2021).

## 1 Introduction

Law Dome is a small ice cap on the coast of East Antarctica (Fig. 1), uniquely positioned to provide exceptionally high temporal resolution ice core records which preserve climate signals from a large sector of the Southern Hemisphere where there are limited historical and instrumental records available. It is situated on a promontory along the Budd Coast of East Antarctica which is bounded by the Totten and Vanderford glaciers, separating it from the main ice flow in the interior of the Antarctic ice sheet. Its climate is driven largely by its position in the path of the large cyclonic weather systems originating from the Southern Ocean (Bromwich, 1988; Udy et al., 2021), which in turn provide a link to lower latitude climate. These cyclonic weather systems result in relatively high annual snowfall with orographic influences resulting in a snowfall gradient across the dome from east to west with higher annual snowfall to the eastern side (Pfitzner, 1980). The high snowfall accumulation rate at the summit is predominantly uniform throughout the year (van Ommen and Morgan, 1997) with well preserved annual layers, allowing for ice core records of annual to seasonal resolution and clear dating which have provided high resolution proxy data of Southern Hemisphere climate over the past 2000 years. In particular, it provides high resolution records of the Pacific and Indian sectors of the Southern Ocean, where there are limited historical or instrumental observations.

Law Dome's close proximity to Casey Station has made it a relatively easy site to access for ice coring activities and repeat site visits to update records with shallow cores. Ice cores have been collected from various sites on Law Dome and have been shown to record regional, hemispheric and global records of past climate over the common era. The ice core from the DSS site provides a long record extending back to the last glacial maximum, but with very high temporal resolution over the past 2000 years. Measurements in cores from elsewhere on Law Dome have included gases trapped within ice to investigate past atmospheric composition (Etheridge et al., 1998; Rubino et al., 2019). Stable isotopes of water from the DSS site show a regular seasonal cycle helping to identify annual layers and enabling the estimation of the annual snow accumulation (Roberts et al., 2015), and have wider uses in southern Indian Ocean variability (Masson-Delmotte et al., 2003) and in large compilations of past global temperature variability reconstruction (eg Ahmed et al. (2013); Emile-Geay et al. (2017)). Proxies derived from annually-dated ice core records from Law Dome have been used to reconstruct climate modes such as El Niño-Southern Oscillation (ENSO) and the Interdecadal Pacific Oscillation (IPO) (Vance et al., 2015) and have been used as records of Australian rainfall (van Ommen and Morgan, 2010; Vance et al., 2013) and streamflow (Tozer et al., 2018), for evaluation of solar and meteorological influences on the $^{10}$Be solar activity proxy (Pedro et al., 2011a, b, 2012), volcanic activity (Plummer et al., 2012) and sea ice extent (Curran et al., 2003).

Palaeoclimate records from East Antarctica provide important longer term context of trends and variability for the shorter instrumental records that are available. The Law Dome record is significant as it is one of the few continually updated sources of palaeoclimate data in this region. Publishing the lastest, version controlled record to ensure the most accurate data are available and citable is important for input into large community climate compilation and synthesis efforts such as the AntClim21 and AntClimNow SCAR initiatives and projects under the Past Global Changes (PAGES) 2k network. Examples of projects with particular reliance on ice core records include reconstructions of climate variability over the last 2000 years by (Stenni et al., 2017), synthesis and reconstruction of snow accumulation patterns across Antarctica (Thomas et al., 2017) and global temperature trends (Emile-Geay et al., 2017).

This collection of records supersedes all previously released versions of data from the Dome Summit South (DSS) site on Law Dome, with the composite record described here forming the best continuous record available, with improved subannual dating. Repeated visits to the DSS site has allowed the record to be continuously updated, extending the overlap with the instrumental data period. Further analysis of previously collected ice core material at finer resolution than previously performed has enabled improvements to the subannual dating.

We provide trace chemistry, stable water isotopes and snow accumulation data, with some higher-level products also available. Level 1 data consists of quality-controlled trace chemistry data for chloride ($Cl^-$), sodium ($Na^+$), magnesium ($Mg^{2+}$), calcium ($Ca^{2+}$), potassium ($K^+$), sulphate ($SO_4^{2-}$), nitrate ($NO_3^-$) ions. At this time level 1 stable water isotope data is not included. Level 2 datasets are derived from the level 1 products, with annual averages provided for each chemical species, $\delta^{18}O$ and snow accumulation as well as seasonal sea salts. These derived datasets serve as examples of what we believe to be the most suitable method for annualising each data stream. We also detail the limitations which should be considered in their application in further research.

Further visits to this site for collecting ice cores is expected as part of an ongoing monitoring project, with subsequent updates to this record when ice cores become available for analysis. These regular visits to update the record by collecting shallow cores extends the period which overlaps with modern satellite observations, ensuring the record stays up to date and more data is available for calibration of palaeoclimate reconstructions.

This publication provides a reference point for the full suite of data, with links to versioned and updated data publicly available for download from the Australian Antarctic Data Centre at https://doi.org/10.26179/5zm0-v192 (Curran et al., 2021). Note that in the text of this manuscript we use years CE for dates as the more natural dating scale, and to be consistent with the naming of some cores using the year or season in which they were drilled. The datasets available for download provide years in both CE and BP 1950 scales and we use BP 1950 (i.e. years before 1950 CE) for plotting of data in figures here.

## 2 Methods

### 2.1 Drilling campaigns

The DSS site is located at 66°46'11"S 112°48'25"E, approximately 4.7 km SSW from the dome summit (Morgan et al., 1997). The drill sites of the four cores included in this composite record are all located within approximately 1 km of each other, with their relative locations shown in Fig. 1(b). Future visits to the site are planned to ensure the record continues to be updated. A summary of the details of the drilling campaigns is found in Table 1, with more detailed description of the original drilling site published by Morgan et al. (1997).

| Core | Drilling period (CE) | Drill type | Diameter (mm) | Length (m) | Co-ordinates |
|------|---------------------|------------|---------------|------------|--------------|
| DSS1617 | 10/02/2017 | Electromechanical (Eclipse) | 80 | 30 | 66° 46'26"S 112° 48'41"E |
| DSS99 | 20/02/00–06/3/00 | Thermal (Horace) | 120 | 125 | 66°46'14"S 112°48'25"E |
| DSS97 | 28/10/97–26/11/97 | Electromechanical (Eclipse) | 82 | 270 | 66°46'38"S 112°48'41"E |
| DSSMain | 1987–93 | Electromechanical (Istuk) | 100 | 1196 | 66°46'11"S 112°48'25"E |

**Table 1.** Details of the drilled cores used in this composite record. Locations of each of the individual cores are also shown in Fig 1(b)

### 2.2 Dating and age horizon uncertainties

The DSS record has been dated using seasonal variations in the deposition of species to define calendar year boundaries, commonly known as annual layer counting. The stable water isotope record is the primary seasonal indicator, with a well-defined maximum of January 10±2.2 days (van Ommen and Morgan, 1997). Confirmation of dating is provided by summer-peaking hydrogen peroxide where available, and other chemical markers, such as summer peaking methanesulfonic acid (MSA), non-sea salt sulphate ($nssSO_4^{2-}$), sulphate/chloride ($SO_4^{2-}/Cl^-$) ratio, and the winter-peaking sea salt species (chloride ($Cl^-$), sodium ($Na^+$), magnesium ($Mg^2+$)). Layer counting of the DSS record between 1300 to 1995 CE was previously completed by Palmer et al. (2001b) and verified and extended to cover over 2000 years by Plummer et al. (2012). The DSS record cur-

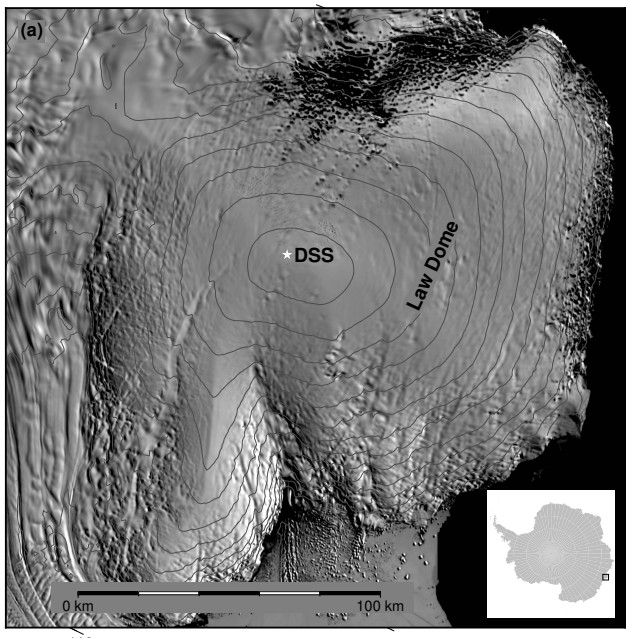
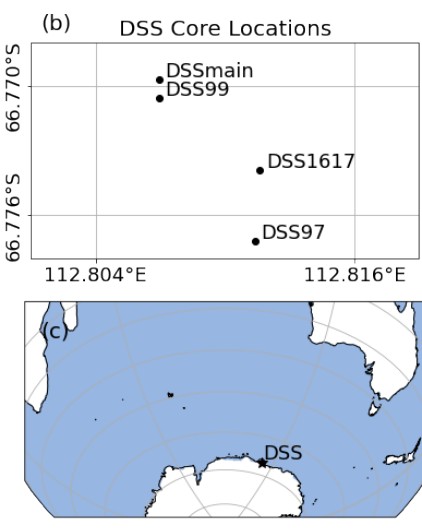

**Figure 1.** (a) Location of DSS ice core site on Law Dome, East Antarctica. Background image is MODIS Mosaic of Antarctica Scambos et al. (2007). (b) Relative locations of the 4 individual drill sites used in this record. (c) Regional context of Law Dome.

rently spans -11 to 2017 CE with the addition of the DSS1617 core which is dated unambiguously from 1989 to 2017 CE. The timescale from Plummer et al. (2012) has been updated by refining the annual layer placement, assisted by new water isotope analysis, and by applying a stricter error counting protocol. The number of years counted has not changed, however the error estimate has increased in the oldest part of the record to +20/-7 at -11 CE, where this date may be up to 20 years older or 7

85   years younger (-31 to -4 CE) as shown in Fig. 2. Where evidence for a year horizon was not clearly identifiable in the primary seasonal indicators, or only weakly supported by confirmatory species, it was not counted. When the majority of (but not all) seasonal indicators show evidence of a year these were counted. Both cases contribute to the total uncertainty estimate. The asymmetrical uncertainty is biased toward under-counting, reflecting increasing likelihood of encountering years with unclear signals with the decreasing layer thickness at depth. DSS is a long record from a high accumulation site ($0.69 \, \mathrm{my}^{-1}$ ice equiv-

alent using an ice density of $917 \, \mathrm{kgm}^{-3}$) and maintains 6-8 chemistry samples per year at -11 CE; enabling development of a long, accurate annual layer-counted chronology. Intra-annual dating of individual sample depths is determined by interpolation between the layer counted annual horizon depths.

    We take a conservative approach to calculating the age horizons and uncertainties. The depth of the top of each 1 m length of drilled core is locked to the top of the bag as recorded in the drilling logs, so uncertainty from the sample resolution is limited

only within each 1 m length and errors in depth arising from missing or badly fragmented core segments are not cumulative. The sample resolution of chemical measurements is generally much greater than the water isotope sample resolution, hence the larger uncertainty value is used.

The DSS chronology presented here has been dated using annual layer counting only; without reference to externally dated reference events (e.g. volcanic events) and is independent of other ice core age scales. There are differences when compared to other ice core chronologies, however those differences are within our error bounds. To assist users to include DSS data in large-scale reconstructions or regional synthesis products, we have provided a translation of the age of the volcanic events used in the PAGES Ant2k compilation (Ahmed et al., 2013) to the WD2014 (Sigl et al., 2016) chronology and our DSS age scale in Appendix A.

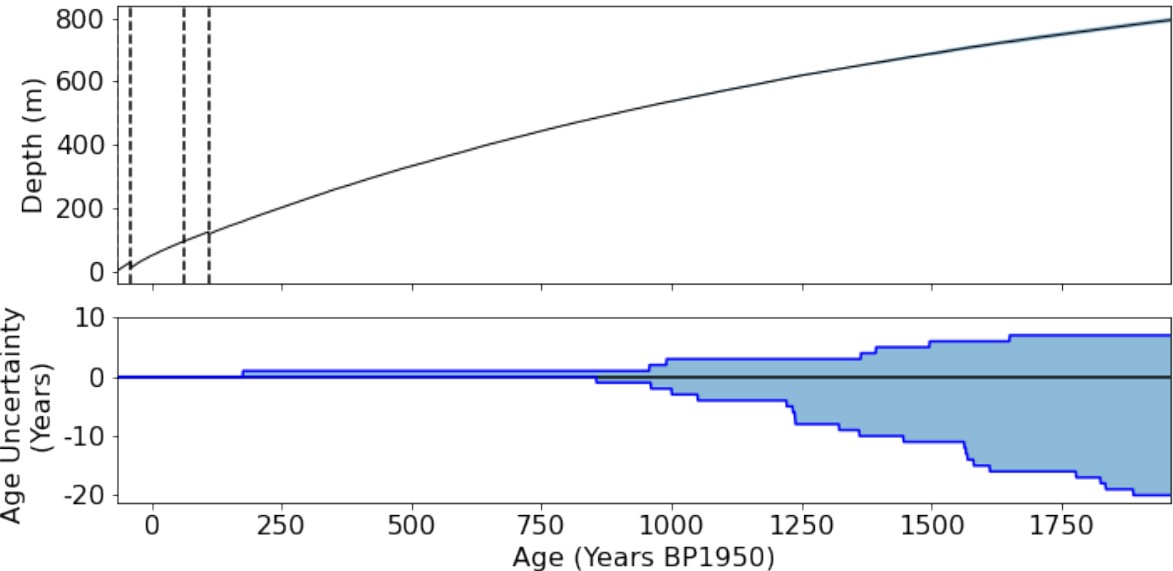

**Figure 2.** (a) Age at depth for the composite record. Steps in the curve correspond to boundaries between cores, as they were drilled years apart. (b) Accumulated age uncertainties over the 2000 years of data.

The depths of the year boundaries for the 2000 years of data, produced using annual layer counting methods as described above, are contained in the file titled "DSS_2k_age_horizons.csv". The column headings for this file are described in Table 2.

| Date (BP 1950) | Date (CE) | Depth (m) | Core name | min error (years) | max error (years) |
|---|---|---|---|---|---|
| -66 | 2016 | 1.82 | DSS1617 | 0 | 0 |
| ⋮ | | | | | ⋮ |
| 1961 | -11 | 793.887 | DSS | 20 | 7 |

**Table 2.** Column headings for the file "DSS_2k_age_horizons.csv" which contains the depths of year boundaries produced by annual layer counting methods. Year horizons by depth are provided along with the specific core used for that year in the compilation. Accumulated minimum and maximum errors in age are provided, calculated using the method described in Section 2.2.

## 3 Level 1 Datasets

The level 1 datasets have undergone calibration, quality control and post-processing of the raw instrument measurements. The datasets here are obtained from four cores (see Table 1), drilled at the Dome Summit South (DSS 66°46' S 112°48' E) at different times. The main 1196 m (DSSMain) core was drilled between 1987 and 1993. The uppermost 117 m of DSSMain was thermally drilled, and the presence of micro-fractures made it unsuitable for trace ion analysis. Two further shallow cores - DSS97 and DSS99 - were drilled to cover this period for improved data quality. The stable water isotope records from all three cores were used to establish and lock unambiguously the dating and overlap between them (Palmer et al., 2001b). The DSS site has been revisited subsequently, with new short cores retrieved to update the DSS record. The most recent update to the DSS record was DSS1617, a 30 m core drilled during the 2016/17 CE austral summer season, covering 1989–2017 CE, providing 7 years of overlap with DSS97. Previous work from DSS has used other composite records using short cores overlapping with DSS97 (Plummer et al., 2012; Vance et al., 2013; Roberts et al., 2015) which are now no longer used as the periods they covered have been superseded by the longer DSS1617. The periods covered by each of these four cores is illustrated in Fig. 3. Where we have a choice of cores, we have typically selected records to avoid the upper most annual cycle of any core (where the friability of surface snow and firn can result in core sections that are more easily compromised during core processing) and to use long, continuous sections from single cores where possible.

The stable water isotope ($\delta^{18}$O) record is sampled at 10–50 mm resolution, with finer physical sampling as annual layers thin with depth. Level 2 annual mean data is provided (see 4.1), however the level 1 datasets of stable water isotopes are not provided at this time but will be published in the future with a manuscript currently in preparation. To assist with dating, sections of cores were measured for hydrogen peroxide at an average resolution of 50 mm as described by van Ommen and Morgan (1996). Discrete chemistry samples were prepared using the clean techniques described by Curran and Palmer (2001). Cores DSS1617, DSS99, DSS97 and DSSMain to 402 m ( 1300 CE) were sampled at an average resolution of 50 mm, providing up to 60 samples per year in the near-surface firn. Due to a sampling error, a section of DSSMain from 251–273 m (1611–1568 CE) was sampled for chemistry at 100 mm resolution, however isotope and peroxide measurements are available at 50 mm resolution to maintain dating integrity. Beyond 402 m, sample resolution changed to 30 mm and below 578 m (979 CE), resolution changed to 25 mm to offset the effects of layer thinning on seasonality.

### 3.1 Trace Ion Chemistry

Ice core samples have been analysed using ion chromatography for chloride ($Cl^-$), sodium ($Na^+$), magnesium ($Mg^{2+}$), calcium ($Ca^{2+}$), potassium ($K^+$), sulphate ($SO_4^{2-}$), nitrate ($NO_3^-$) and methanesulfonic acid ($MSA^-$). Analytical methods have been modified and updated over the course of the analysis, and are summarised as follows (see also Table 3). DSSMain from 117–402 m was analysed using the methods of Buck et al. (1992) at the University of New Hampshire. DSS99, DSS97, and DSSMain from 402–748 m were analysed according to the methods of Curran and Palmer (2001) with a deeper section from 748-794 m using the methods of Plummer et al. (2012). The most recent analysis, DSS1617, was analysed according to Plummer et al. (2012) with the exception of the cation analytical column being changed to an Ionpac CS19 for improved detection

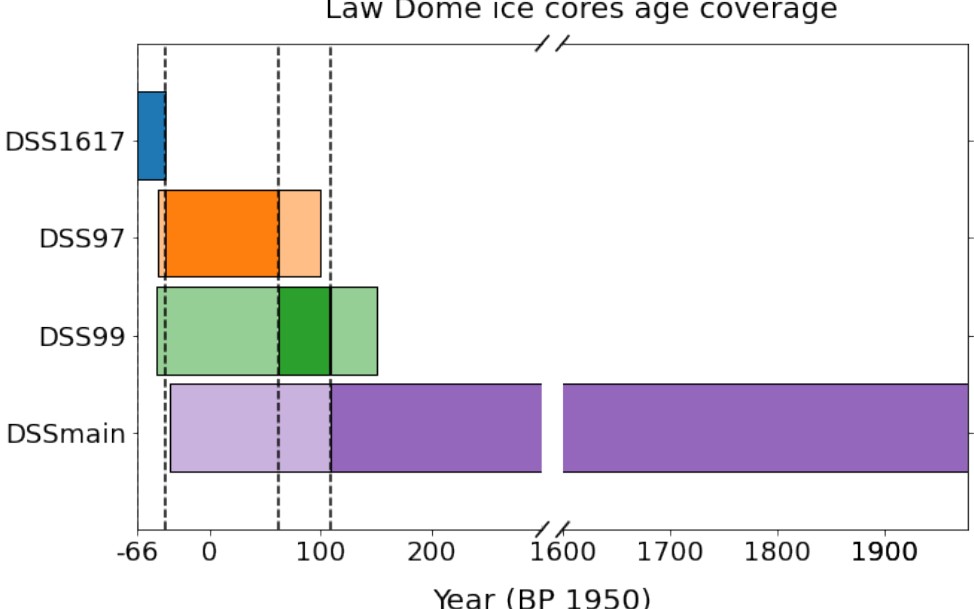

**Figure 3.** Periods of time covered by the individual cores making up this composite record, drilled at DSS from the drilling campaigns, as described in 1. For each individual core, the solid colour indicates the years where data from the core is included in this composite record. Transparent colours indicate where data overlapping the other cores exists, but is not included in the compilation where due to better quality core or data being available.

and peak resolution of magnesium and calcium. Comparison of the analysis methods of Curran and Palmer (2001) and Buck
et al. (1992) were discussed by Palmer et al. (2001b), and showed no significant difference. A comparison of the technique used by Plummer et al. (2012) and Curran and Palmer (2001) similarly found no significant differences.

| Core | Depth range | Average sample resolution | Reference for method used |
|---|---|---|---|
| DSS1617 | whole core | 50 mm | Plummer et al. (2012) + Ionpac CS19 |
| DSS97 | whole core | 50 mm | Curran and Palmer (2001) |
| DSS99 | whole core | 50 mm | Curran and Palmer (2001) |
| DSSMain | 117–402 | 50 mm | Buck et al. (1992) |
| | 251–273 m | 100 mm[†] | Buck et al. (1992) |
| | 402–527 m | 30 mm | Curran and Palmer (2001) |
| | 527–748 m | 25 mm | Curran and Palmer (2001) |
| | 748–794 m | 25 mm | Plummer et al. (2012) |

**Table 3.** Resolution and methods used for trace chemistry analysis on the four different cores used in the compilation. †A sampling error resulted in the larger sample size in this section.

A number of species that were measured on parts of the core are not included in this dataset. MSA was not measured on a significant portion of the record. Additionally, deeper sections of the MSA record suffer from unquantified losses from storage prior to analysis (Roberts et al., 2009). The potassium and calcium records are incomplete and suffer issues with poor detection in some older analyses. Additionally, some sections of the DSSMain calcium record appear to have been affected by dust from storage in a concrete floored freezer. Due to these quality concerns, the potassium and calcium records are not discussed further.

For the remaining datasets we provide for each chemical species: the concentration for each sample used in the composite record, an ID corresponding to the drilled core it was obtained from, the top and bottom depths of each sample and the corresponding age. Summary statistics for the level 1 trace chemistry species are included in Table 4, with the corresponding histograms shown for each species shown in Fig. 4. These show that the the distributions of these species are generally non-normal. In further analysis (such as shown later in this paper in 4.3) the concentrations are log-transformed to partially compensate for the long-tail of the distributions.

| Species | Mean ($\mu$EqL$^{-1}$) | Variance (($\mu$EqL$^{-1}$)$^2$) | Skewness | Kurtosis |
|---|---|---|---|---|
| chloride | 4.25 | 13.26 | 3.06 | 25 |
| nitrate | 0.37 | 0.06 | 7.37 | 145.66 |
| sulphate | 0.77 | 0.17 | 3.17 | 25.11 |
| sodium | 3.58 | 10.98 | 3.50 | 35.06 |
| magnesium | 0.83 | 0.45 | 2.24 | 11.82 |
| non-sea-salt sulphate | 0.46 | 0.16 | 2.58 | 29.13 |

**Table 4.** Summary statistics for the level 1 trace ion chemistry data for each analysed species, calculated over the for the full 2000 years of data included in this compilation.

The level 1 trace chemistry dataset provides a depth and age for the top and bottom of each sample as well as concentrations of the measured ions. These are contained in the file titled "DSS_2k_chemistry_level1.csv", with column headings provided for reference here in Table 5.

| Sample ID | Top depth (m) | Bottom depth (m) | Top Date (BP 1950) | Bottom Date (BP 1950) | Mid Depth Date (BP 1950) | $Cl^-$ ($\mu EqL^{-1}$) |
| --- | --- | --- | --- | --- | --- | --- |
| DSSp1617A-1_9 | 0.255 | 0.300 | -66.981 | -66.953 | -66.967 | 3.967 |
| ... | | | | | | ... |
| DSS 832-6 | 793.863 | 793.888 | 1960.867 | 1961.005 | 1960.936 | 1.063 |

| Sample ID | $NO_3^-$ ($\mu EqL^{-1}$) | $SO_4^{2+}$ ($\mu EqL^{-1}$) | $Na^+$ ($\mu EqL^{-1}$) | $Mg^{2+}$ ($\mu EqL^{-1}$) | $nssSO_4^{2+}$ ($\mu EqL^{-1}$) |
| --- | --- | --- | --- | --- | --- |
| DSSp1617A-1_9 | 0.427 | 1.572 | 3.332 | 0.679 | 1.281 |
| ... | | | | | ... |
| DSS 832-6 | 0.203 | 0.598 | 1.195 | 0.149 | 0.494 |

**Table 5.** Column headings Level 1 trace ion chemistry data file.

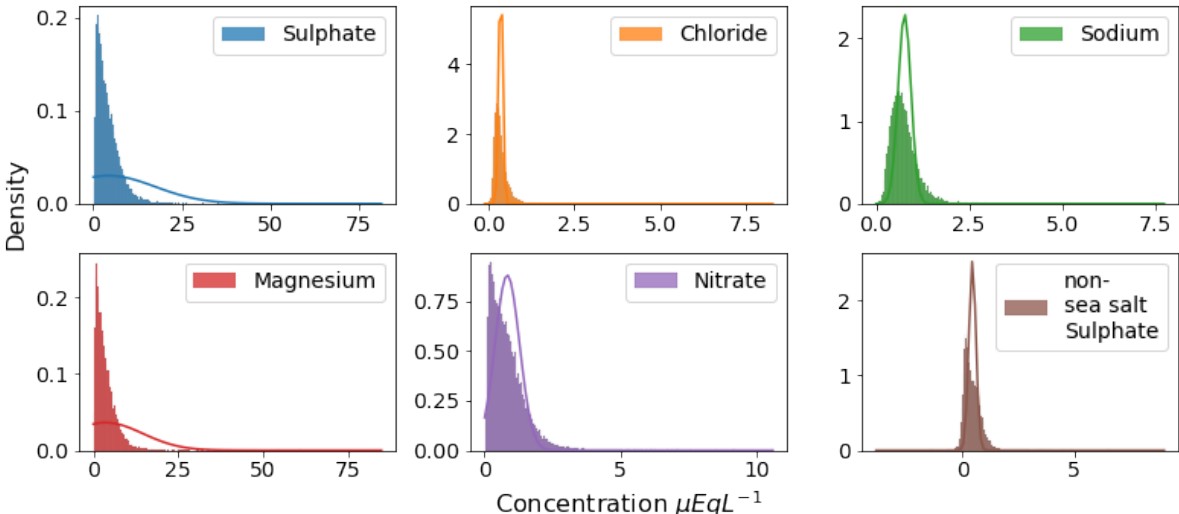

**Figure 4.** Histogram of the distributions of the level 1 trace chemistry data. Histogram of the concentration of each chemical species, with Normal probability distribution function overlaid, illustrating the non-Gaussian distribution of several of the analytes.

Further statistical analysis was performed to examine the consistency in the chemistry data obtained from the different cores within the composite, even where there is little or no overlapping data. As the concentrations of trace ions in the ice may depend on the accumulation rate, we have analysed the cores for epochs of approximately constant accumulation rate, and only compared sections which have similar (but not identical) accumulation rates. The detailed results of this analysis show that the distributions of the chemical species concentrations in the different cores are similar in epochs of comparable snow accumulation rate. The results and details of the analysis performed is found in Appendix B.

## 3.2 Density

The firn densification in this region of Law Dome was established from density measurements taken from several cores in the vicinity of DSS. Samples were prepared from dry drilled cores or from melt free interior of thermally drilled cores and were either machined to cylinders on a lathe or pressed (cookie cutter) from softer firn to a precise volume.

An empirical fit to the firn density profile with depth was presented in van Ommen et al. (1999). Specifically the density ($\rho$, in $\mathrm{kg\,m^{-3}}$) is fitted as a compound function of depth ($z$, in m) as

$$\rho(z) = 917 - 147.55 \exp\left(-\frac{z}{7.1504}\right) + \begin{cases} 4.2826z - 371.56 & \text{if } z < 56.972 \\ -648.65 \exp\left(-\frac{z}{35.034}\right) & \text{if } z \geq 56.972 \end{cases} \tag{1}$$

The density observations and empirical fit as a function of depth (see Eqn. 1) are plotted in Fig. 5, showing good agreement between the two.

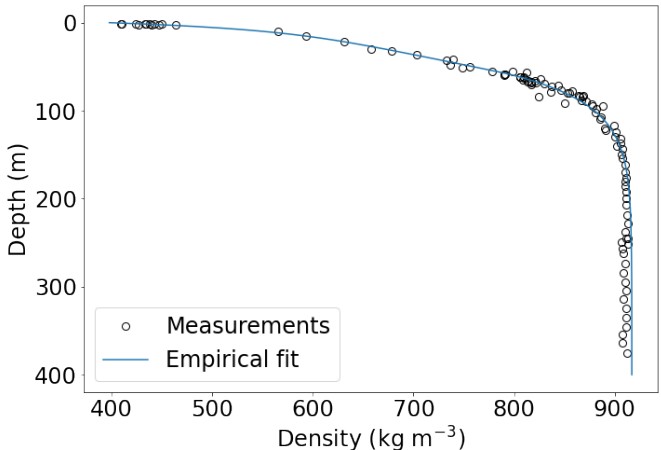

**Figure 5.** DSS density profile, observations (circles) and empirical fit (solid line, calculated using Equation 1).

### Ice Equivalent Depths

For some derived data products (e.g. snow accumulation history) it is useful to be able to remove the effects of firn densification. One common method is to utilise "ice equivalent depths" that represent the depth of a column numerically compressed to the
same density as glacial ice (density $\rho_{ice} = 917 \, \mathrm{kg \, m^{-3}}$) and same cross-sectional area, with the same mass as the firn column. Algebraically, the ice equivalent depth ($\mathcal{Z}$) is given by

$$\mathcal{Z}(z) = \int_0^z \frac{\rho(\eta)}{\rho_{ice}} d\eta = \begin{cases} 0.5948z + 0.002335z^2 + 1.1505 \exp(-z*0.1399) - 1.1505 & \text{if } z < 56.972 \\ z + 1.1505 \exp(-0.1399z) + 24.7817 \exp(-0.02854z) - 21.5316 & \text{if } z \geq 56.972 \end{cases} \quad (2)$$

## 4 Level 2 datasets

The level 2 datasets are derived from the level 1 datasets above. Those presented here have largely been previously published or
included in large data compilations but are now updated, with improved dating of the core. Where there are differences between the versions, these are remarked upon here. These level 2 derived datasets can be considered as examples of best practise for utilising the level 1 data. A plot of the timeseries for all level 2 datasets is shown in Fig. 6.

Two CSV format text files are provided for the Level 2 data sets, "DSS_2k_winter_centred.csv" and "DSS_2k_summer_centred.csv" which contain the annually averaged datasets as described in Section 4. Column headings are provided for reference in Tables 6
and 7. These datasets are provided together to ensure that a consistent age scale across all the derived datasets.

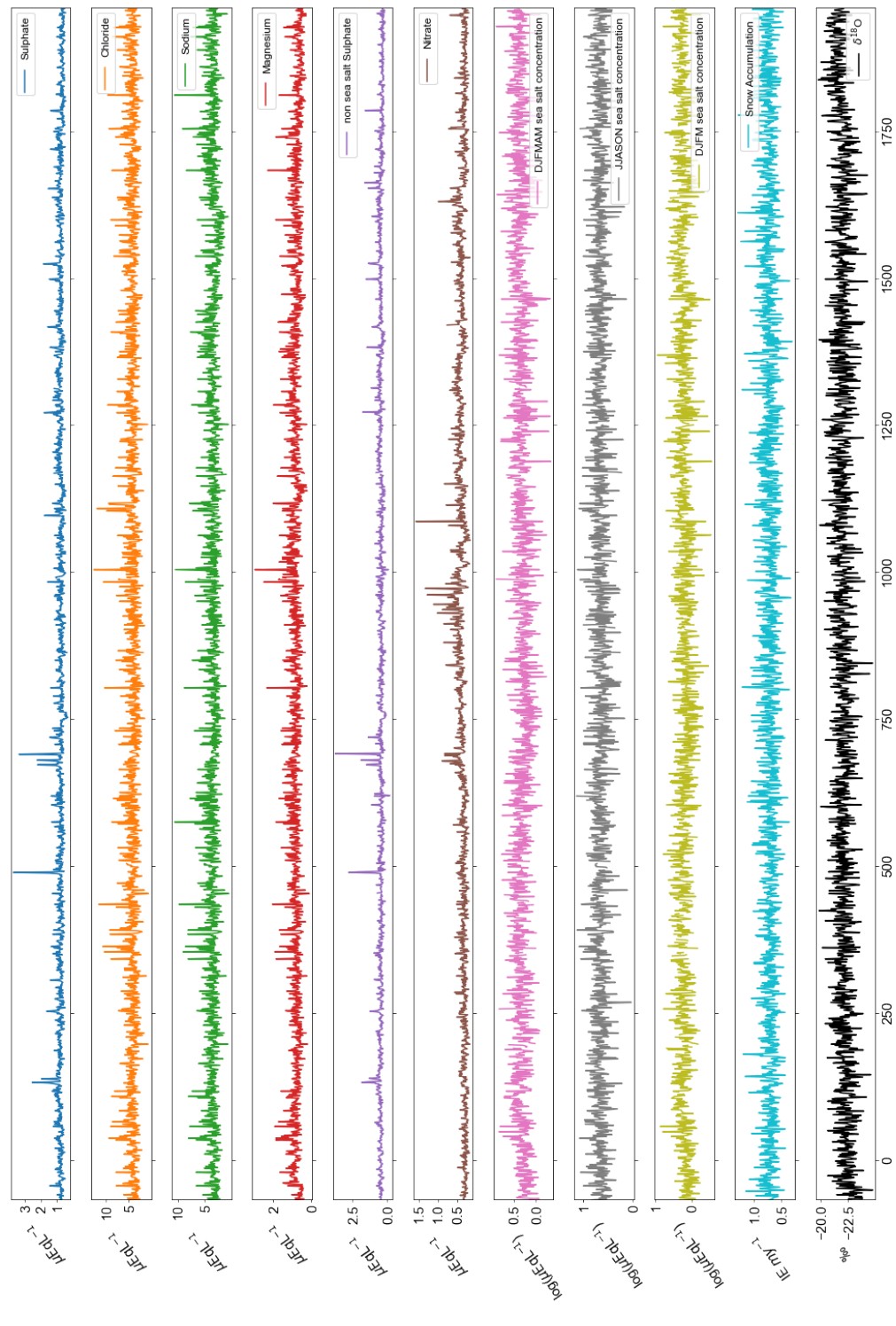

**Figure 6.** Time series for all annual records, include trace chemistry analytes, stable water isotopes, accumulation and derived seasonal sea salts.

| Year (BP 1950) | Year (CE) | Na $(\mu EqL^{-1})$ | Cl $(\mu EqL^{-1})$ | Mg $(\mu EqL^{-1})$ | $SO_4$ |
|---|---|---|---|---|---|
| -66 | 2016 | 4.326 | 4.994 | 0.844 | 0.795 |
| $\vdots$ | | | | | $\vdots$ |
| 1961 | -11 | 2.925 | 3.687 | 0.46 | 0.589 |

| Year (BP1950) | Year (CE) | $\delta^{18}O$ (‰) | Layer Thickness (m) | Accumulation rate $(IE\,m\,y^{-1})$ |
|---|---|---|---|---|
| -66 | 2016 | -22.511632 | 0.733355 | 0.734644 |
| $\vdots$ | | | | $\vdots$ |
| 1961 | -11 | -22.103803 | 0.18 | 0.662081 |

| Year (BP1950) | Year (CE) | DJFMAM $\left(\log\left(\mu EqL^{-1}\right)\right)$ | JJASON $\left(\log\left(\mu EqL^{-1}\right)\right)$ | DJFM $\left(\log\left(\mu EqL^{-1}\right)\right)$ |
|---|---|---|---|---|
| -66 | 2016 | 0.154 | 0.83 | 0.338652 |
| $\vdots$ | | | | $\vdots$ |
| 1960 | -10 | NaN | 0.543 | NaN |
| 1961 | -11 | NaN | NaN | NaN |

**Table 6.** Winter centred annual data columns included in "DSS_2k_winter_centred.csv" file.

| Year (BP 1950) | Year (CE) | $NO_3^-$ $(\mu EqL^{-1})$ | $nssSO_4^{2+}$ $(\mu EqL^{-1})$ |
|---|---|---|---|
| -66.5 | 2016.5 | NaN | NaN |
| -65.5 | 2015.5 | 0.31 | 0.453 |
| $\vdots$ | | $\vdots$ | |
| 1959.5 | -9.5 | 0.363 | 0.33 |
| 1960.5 | -10.5 | NaN | NaN |

**Table 7.** Summer centred annual data columns included in "DSS_2k_summer_centred.csv" file.

## 4.1 Annual stable water isotopes

The DSS annual stable water isotope presented here updates and extends the stable isotope record included in Emile-Geay et al. (2017) and Stenni et al. (2017) using the DSS1617, DSS99, DSS97 and DSSMain cores (see Fig. 3), and currently spans the period -11 to 2016 CE. The previously published annual composite record for DSS, used in Antarctic temperature reconstructions (Ahmed et al., 2013; Emile-Geay et al., 2017; Stenni et al., 2017), included other short cores which are now no longer used as the time periods they covered are now superseded by the longer DSS1617. Also, further analysis has extended the annual stable water isotope for the DSSMain record to cover -11 to 174 CE. The record presented here includes some changes from the previous records due to the new cores, minor changes in the year boundaries and the new analysis data.

Annual averages are calculated using the year boundaries defined by the summer peak in $\delta^{18}$O. Changes from previously
released records have occurred due to new data being obtained from the DSS1617 core as well as new analysis at depth from
the original DSSMain core. Minor adjustments have been made to the year/depth horizons and corrections to core flips, sample
mishandling and depth errors which have been identified since the last release and have been verified by the chemistry records.
The time series for the annual averaged $\delta^{18}$O data is shown in Fig. 7.

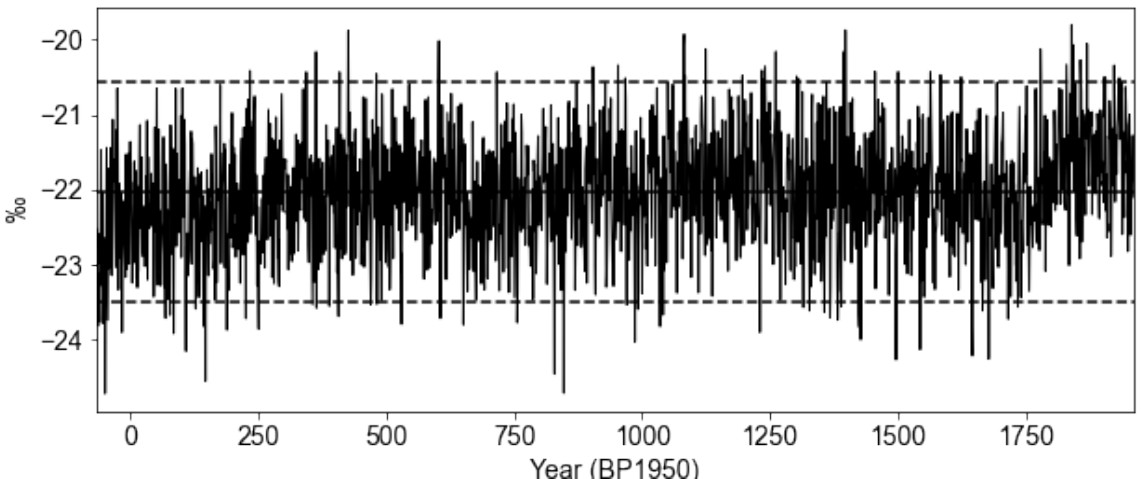

**Figure 7.** Time series for annually averaged $\delta^{18}$O. Solid black line indicates the 2000 year mean value. Dashed black lines indicates $\pm 2\sigma$ values.

## 4.2 Annual trace ions

Annual average trace ion concentration values are presented here, and the calculation method is species dependent. A winter-centred average is used for winter-peaking species (e.g. sea salts) with the boundaries set at the beginning and end of each calendar year. The summer-centred value is calculated from mid-year to mid-year, and used for the nominally summer-peaking species nitrate and non-sea-salt sulphate. This was done to reduce edge effects where small differences in year boundary placement could have strong effects on summer-peaking species. The non-sea-salt sulphate concentration is calculated according to the method of Palmer et al. (2001a) and has a fractionation correction applied to minimize non-physical negative values. The distributions of the annually averaged records is shown in Fig. 8.

Plots of the annual average time series for each of the measured chemistry species are given below, along with the median value, $2\sigma$ threshold and the number of samples used for each annual average shown in the lower panel (see Fig.s 9 – 14).

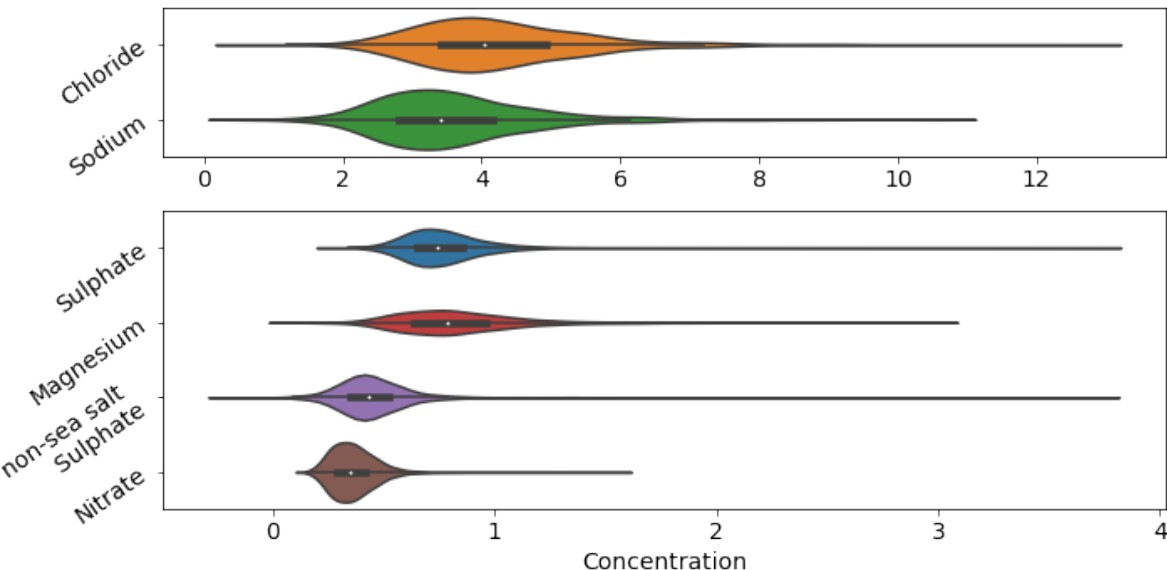

**Figure 8.** Violin plots showing the median, interquartile range and distribution of the annually averaged trace ions. The chloride and sodium records are separated only for illustrative purposes so that their higher concentrations do not dominate the plots of the other ions.

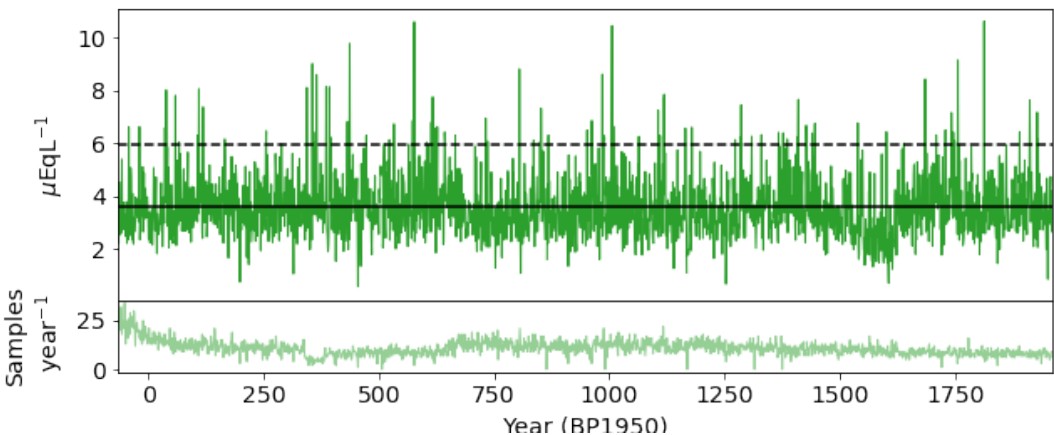

**Figure 9.** Time series for annually averaged sodium. Solid black line indicates the 2000 year mean value. Dashed black line indicates $2\sigma$ value. The lower panel indicates the number of individual samples in the year used for the average value.

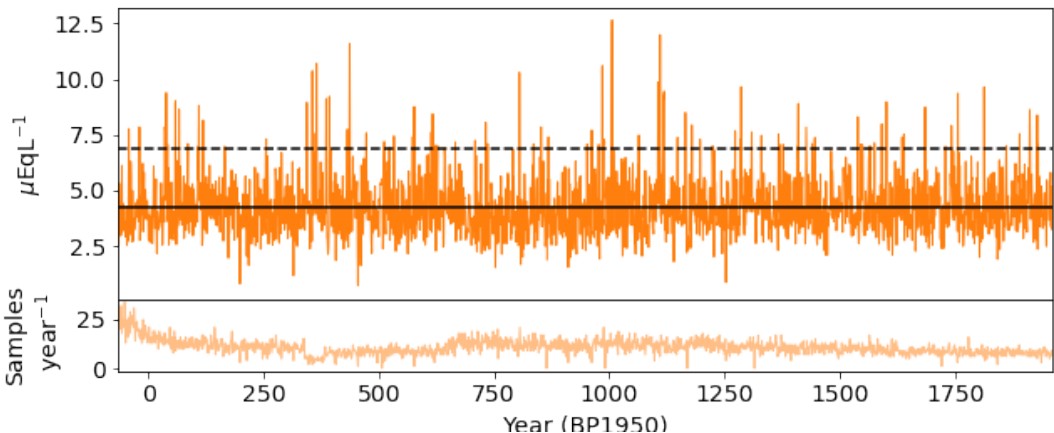

**Figure 10.** Time series for annually averaged chloride. Solid black line indicates the 2000 year mean value. Dashed black line indicates $2\sigma$ value. The lower panel indicates the number of individual samples in the year used for the average value.

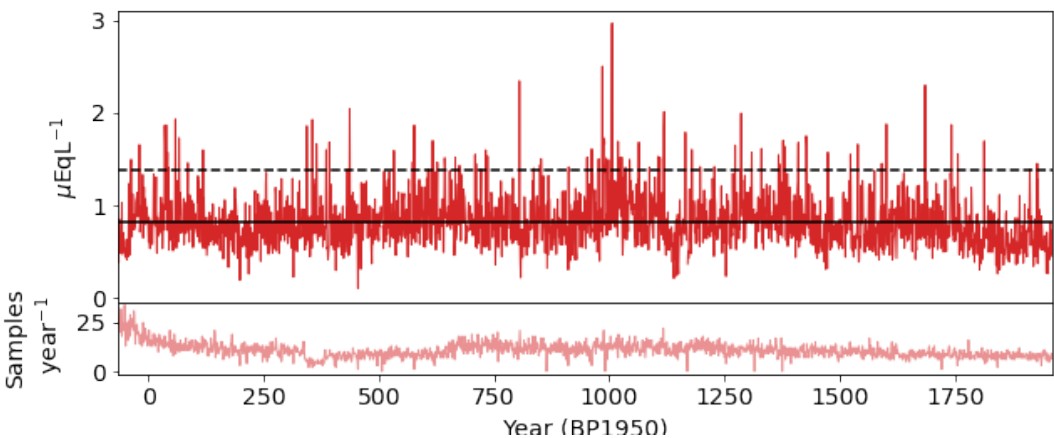

**Figure 11.** Time series for annually averaged magnesium. Solid black line indicates the 2000 year mean value. Dashed black line indicates $2\sigma$ value. The lower panel indicates the number of individual samples in the year used for the average value.

## 4.3 Seasonal sea salts

Characteristics of seasonal cycles of the different species of ions from the Law Dome ice core has been studied over short time periods (Curran et al., 1998), while annual sea salt data has been used previously for high-precision dating of volcanic events (Palmer et al., 2001b), and used for palaeoclimate studies of atmospheric circulation (Souney et al., 2002).

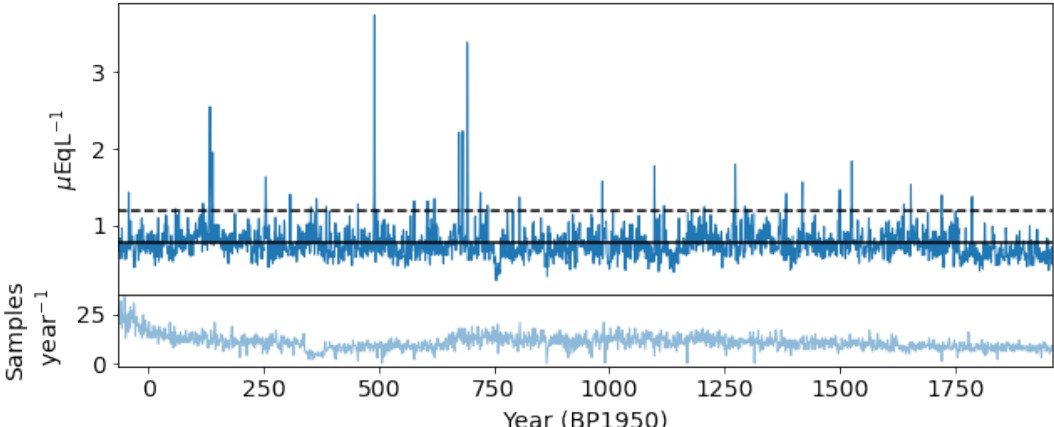

**Figure 12.** Time series for annually averaged sulphate. Solid black line indicates the 2000 year mean value. Dashed black line indicates $2\sigma$ value. The lower panel indicates the number of individual samples in the year used for the average value.

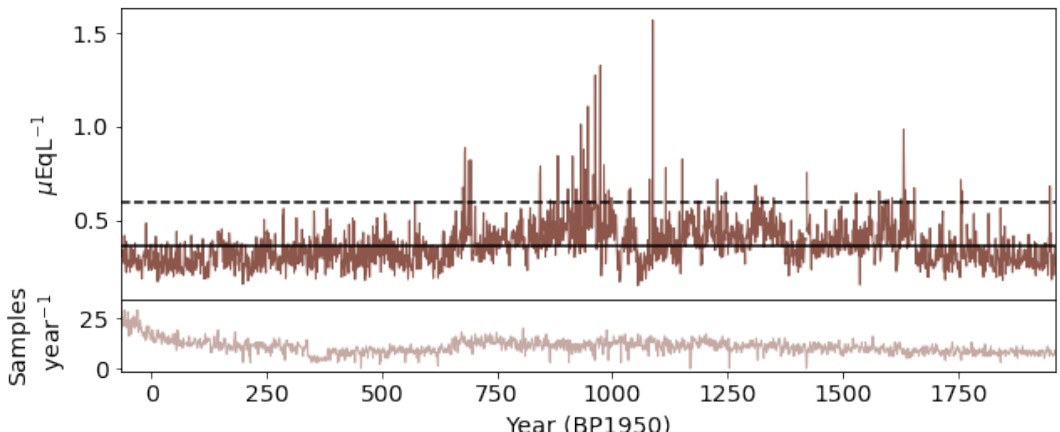

**Figure 13.** Time series for annually averaged nitrate. Solid black line indicates the 2000 year mean value. Dashed black line indicates $2\sigma$ value. The lower panel indicates the number of individual samples in the year used for the average value.

Three seasonal sea salt records were developed from aggregating the level 1 sodium record for climate proxy and reconstruction purposes in Vance et al. (2013, 2015) and further used Crockart et al. (2021); Vance et al. (2021). Specifically these three records include a prescribed summer season of December to March inclusive mean chloride concentrations, hereafter the 'summer sea salt record', as well as a semi-annual aggregation into a defined warm (December to June inclusive) and cool (July to November inclusive) mean salt concentration. The summer sea salt record was developed as a proxy for rainfall variability in eastern Australia as well as a Western Tropical Pacific ENSO proxy over the Common Era, and is provided as log-transformed

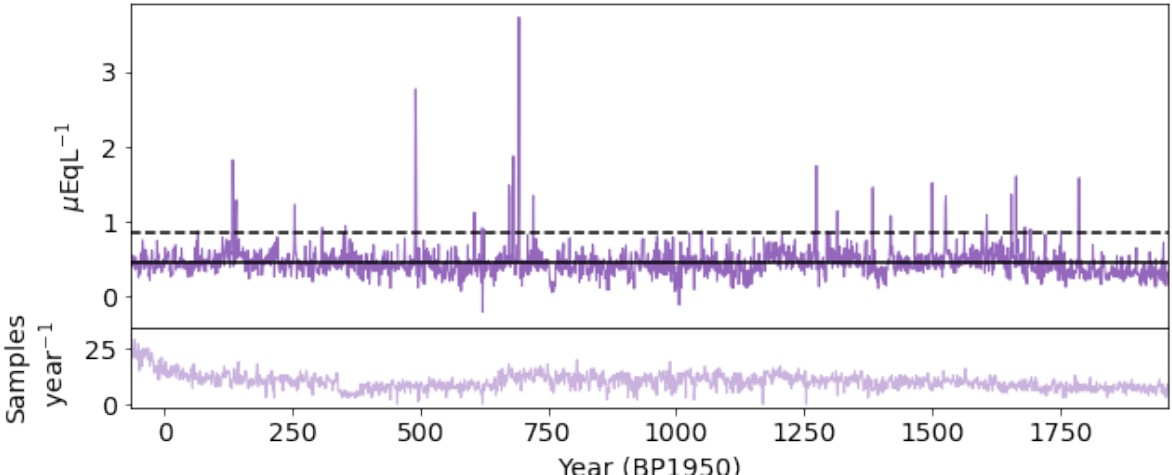

**Figure 14.** Time series for annually averaged non-sea salt sulphate.Solid black line indicates the 2000 year mean value. Dashed black line indicates $2\sigma$ value.

mean concentrations. Variability in summer sea salt concentrations is attributed to synoptic scale changes in the southern Indian
Ocean (Udy et al., 2021). The warm and cool season mean sea salt concentrations were similarly log-transformed and provided input timeseries for reconstructing Pacific Ocean decadal variability, specifically the Interdecadal Pacific Oscillation index (Parker et al., 2007; Vance et al., 2015, 2016). These datasets have been extended and undergone improved quality control procedures, as detailed in Vance et al. (2021) and adopted here, which extended the IPO reconstruction to span the Common Era.

These seasonal records were derived by binning the quality controlled, dated sea salt data from the level 1 datasets into 12 months per year. The salt concentrations were log-transformed to normalize (as sea salt data from Law Dome displays long tails due to high frequency aerosol generation events from synoptic activity in the southern Indian Ocean). After binning, timeseries of seasonal averages (December–March, December–May and June–November) were produced with the following caveats. The level 1 sea salt records were examined in conjunction with the stable water isotope records and field and laboratory logbooks
detailing core morphology and ice sample cutting. In some instances, the binned sea salt data was compromised due to either missing ice core material from core breaks or shattered sections, or analytical problems resulting in no data or suspected contamination. If this had occurred, the data in question was removed from the analysis. Where more than one month of data was compromised for the summer sea salt record, or two months for the warm and cool season records, we elected to not include that season in our published timeseries. This results in missing values in the seasonal sea salt timeseries of summer,
warm and cool season salt concentrations over the last 2000 years and are recorded as NaN in the dataset files. Timeseries for the summer, warm and cool season log-transformed sea salts are shown in Figs. 15, 16 and 17.

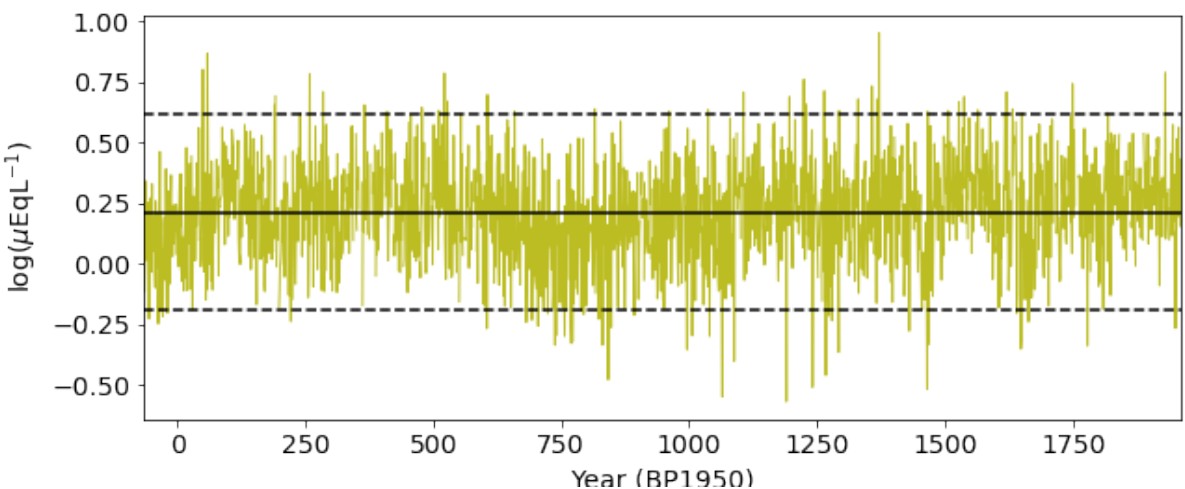

**Figure 15.** Time series for DJFM (Summer) annual sea salts concentration. Solid black line indicates the 2000 year mean value. Dashed black line indicates $\pm 2\sigma$ value.

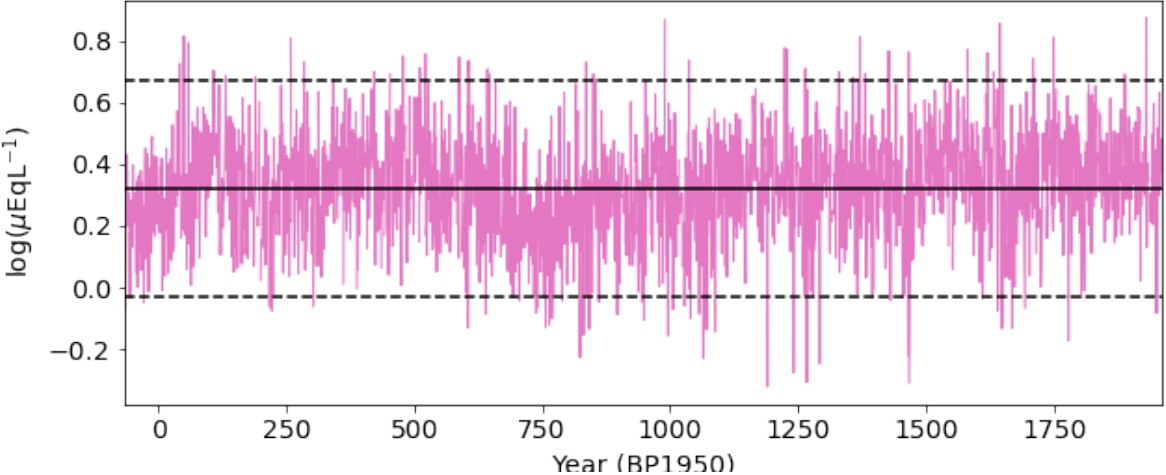

**Figure 16.** Time series for DJFMAM (warm season) annual sea salts concentration. Solid black line indicates the 2000 year mean value. Dashed black lines indicates $\pm 2\sigma$ values.

## 4.4 Annual snow accumulation

Snow accumulation is derived using the year horizons obtained through the layer-counted dating. This record has been previously published in Roberts et al. (2015) and included in PAGES 2k snow accumulation data compilation (Thomas et al., 2017).

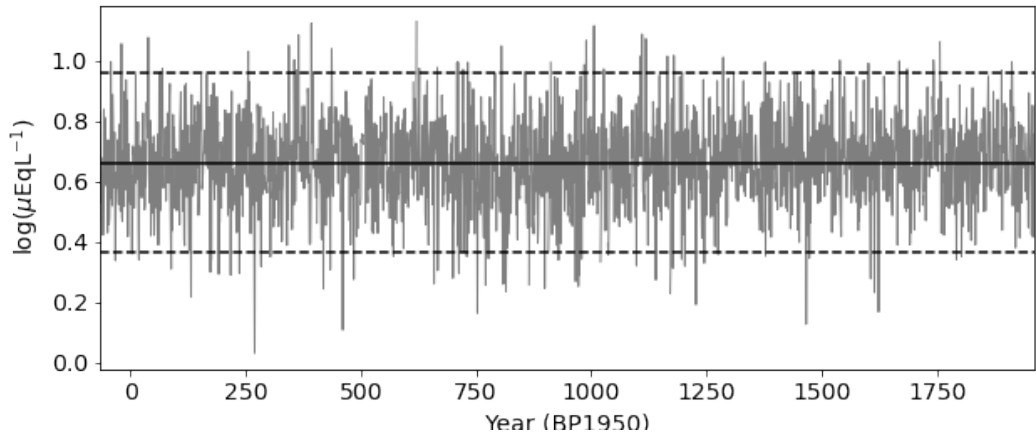

**Figure 17.** Time series for JJASON (cool season) annual sea salts concentration. Solid black line indicates the 2000 year mean value. Dashed black lines indicates $\pm 2\sigma$ values.

This update includes the newer DSS1617 core data and the improved dating. We assume steady state for depth profiles of both density (Sorge's law) and the vertical strain rate. To compensate for firn densification effects, the depths for year boundaries have been converted to ice equivalent depths using Equation 2. Annual snow accumulation rates were then estimated using the power-law vertical strain rate method of Roberts et al. (2015), based on the assumption of no long-term trend in snow accumulation rate. The resulting snow accumulation rate is shown in Fig. 18, with the long term mean snow accumulation rate

of $0.691\pm0.004$ IE m y$^{-1}$.

## 5 Summary

In this paper we have presented the suite of data covering the last 2000 years from the Law Dome ice cores. This set has been updated from previously published results with improved and extended annual dating. We provide quality controlled trace ion chemistry datasets as well as derived, annualised products.

This dataset can be used for high resolution studies of southern hemisphere climate drivers at annual to seasonal scales

    Our aim for this paper has been to provide the community with the most complete and consistent suite of Law Dome for the past 2000 years, and examples of best-practise methods for generating derived datasets such as the annual means. The data sets here supersede all previous releases. This dataset is useful for future high resolution studies of palaeoclimate proxies as has been useful in the past. Future users of the data should take the following summary of the points into consideration

1. The dating presented has uncertainties due to occasionally unclear seasonal markers. The dating is biased towards undercounting of years with unclear boundaries.

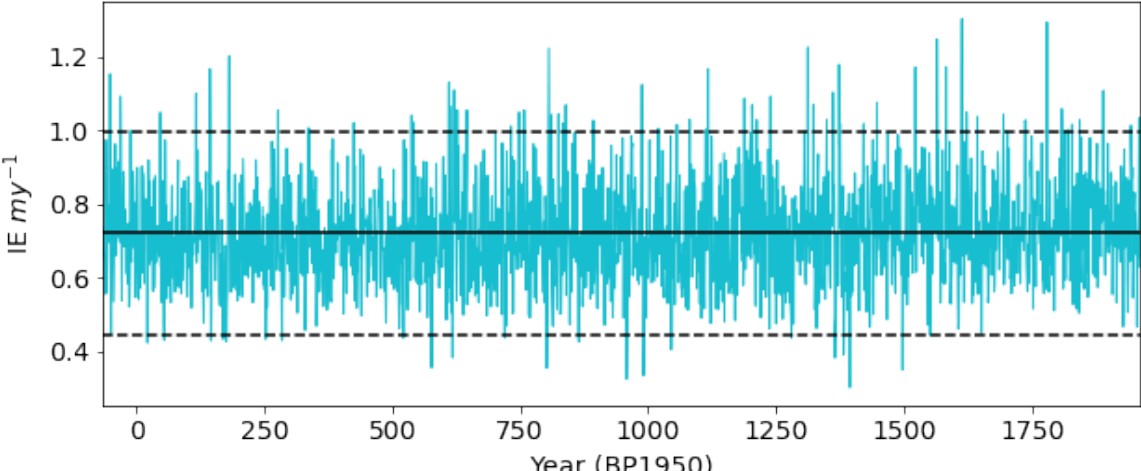

**Figure 18.** Time series for the annual snow accumulation rate. Solid black line indicates the 2000 year mean value. Dashed black line indicates $\pm 2\sigma$ value.

2. As this dataset is dated through annual layer counting, independent of reference to volcanic or solar activity ties, there is an offset in the chronology compared to other ice core chronologies. A translation in ages of volcanic ties is provided to facilitate the inclusion of this data where a common age scale is required.

3. While we provide all the level 1 chemistry data, it is more useful to consider annualised values for the data, either summer or winter centered as we have described depending on the particular analyte. Future users of the data remain free to define seasonal boundaries as best suits their particular needs.

4. Seasonal averages are possible, as provided here in the recently published Vance et al. (2021) as an example, it should be noted that this data has undergone further quality controls resulting in no average value being recorded where there
are an insufficient number of clear records within the seasonal bounds to calculate an average with confidence.

We encourage all potential users of the data to contact us if they have any further queries about how best to use the data.

## 6  Data availability

The dataset described in this manuscript can be accessed at the Australian Antarctic Data Centre under https://doi.org/10.26179/5zm0-v192 (Curran et al., 2021). Individual files may be accessed by clicking on "View Dataset Contents".

*Author contributions.*  All the authors were involved in the generation of the dataset, through collection, analysis, quality control and dating. LMJ co-ordinated the compilation of the data and led the writing of the manuscript with CP and contributions from all the other authors.

*Competing interests.* The authors declare that they have no conflict of interest.

*Acknowledgements.* The authors with to acknowledge the contribution of the expeditioners involved in drilling the ice cores and laboratory support staff. This research has been supported by the Australian Government Department of Industry, Science, Energy and Resources (grant no. ASCI000002) the US National Science Foundation (OPP-9811857 and ATM-9808963), the Australian Research Council Discovery Project DP180102522, Australian Research Council Special Research Initiative for Antarctic Gateway Partnership SR140300001 and past funding from the Antarctic Climate & Ecosystems Cooperative Research Centre (ACE CRC, 2010-2019). The Australian Antarctic Division provided funding and logistical support (ASS project 757 and AAS projects, 4061, 4062, 4537).

## Appendix A: Translation to other age scales

The Law Dome record remains independently dated using annual layer counting techniques, without reference to tie points in other ice core chronologies. As users of this data may wish to map the Law Dome data onto other commonly used ice core chronologies, we provide here a mapping of the age of volcanic tie points between the Law Dome age scale to the Ant2k (Ahmed et al., 2013), the AICC2012 (Veres et al., 2013)) and WD2014 (Sigl et al., 2016) age scales. As the Law Dome record from DSS is a composite record consisting of cores drilled at different times over a period of 4 decades it is difficult to provide a depth for each year using these chronologies, so we provide a "year-to-year" mapping of the volcanic tie points across these age scales.

## Appendix B: Trace Chemistry statistical analysis

Due to only very short overlaps between the various ice cores, there is insufficient data for a reliable direct assessment of the consistency of chemical species between the cores. Therefore, as an alternative, we identify different epochs when we might expect similar distributions, specifically periods of time covered by one of the ice cores where the average accumulation rate approximately the same in a different core making up the compilation but during a different period of time. Changepoint analysis was performed on the cumulative summation (CUSUM) of the snow accumulation record to determine different epochs of approximately constant accumulation, a technique that has been used to detect changes in the Law Dome snowfall data by Zheng et al. (2021). The CUSUM simply sums the data anomalies and identifies a step change in the underlying data through changes in the gradient of the CUSUM. The changepoints in the CUSUM gradient were detected using the Ruptures Python package (Truong et al., 2020). We select epochs of the closest mean accumulation value and then compare between the different cores using a Kolmogorov-Smirnov (K-S) and a Welch's t-test (W-T) on the log transformed data for each of chemical species. These two tests provide two different ways to test the populations, with the K-S test not assuming a normally distributed population, while log transforming the data for the W-T test mitigates some of the long tail of the distribution. The epochs that were compared are shown in Fig. A1, with a summary of the statistical results in Table A1. Visual inspection was also performed using the empirical cumulative probability distributions of the log transformed concentration data, in Fig. A2.

| Ant2k date (CE) | NGRIP depth (m) | AICC2012 Date (NGRIP) | WAIS depth (m) | WAIS 2014 (CE) | DSS depth (m) | DSS 2022 (CE) |
|---|---|---|---|---|---|---|
| 1992 | | | 6.68 | 1991.95 | 28.55† | 1992.08 |
| 1885 | 35.59 | 1884.82 | 40.71 | 1885.01 | 97.15‡ | 1884.99 |
| 1836 | 47.86 | 1836.23 | 53.99 | 1835.10 | 120.53 | 1835.91 |
| 1816 | 52.42 | 1816.84 | 59.06 | 1815.87 | 133.48 | 1816.16 |
| 1811 | 53.68 | 1811.13 | 60.40 | 1810.76 | 136.78 | 1811.05 |
| 1696 | 78.33 | 1696.21 | 87.81 | 1695.90 | 203.92 | 1696.01 |
| 1642 | 89.27 | 1641.71 | 100.52 | 1642.21 | 234.69 | 1642.09 |
| 1602 | 97.07 | 1601.73 | 110.25 | 1600.95 | 257.45 | 1600.93 |
| 1595 | | | 111.70 | 1594.90 | 260.86 | 1594.43 |
| 1460 | 124.11 | 1459.80 | 142.96 | 1459.74 | 328.30 | 1459.57 |
| 1346 | 145.90 | 1345.63 | 168.89 | 1346.55 | 380.75 | 1345.03 |
| 1278 | 158.16 | 1277.70 | 184.59 | 1277.25 | 411.38 | 1276.68 |
| 1270 | 159.46 | 1270.56 | 186.19 | 1269.73 | 414.65 | 1268.88 |
| 1259 | 161.49 | 1259.53 | 188.70 | 1258.87 | 418.89 | 1258.27 |
| 1242 | | | 192.79 | 1241.88 | 425.97 | 1241.28 |
| 1231 | 166.67 | 1230.64 | 195.34 | 1230.68 | 430.61 | 1229.93 |
| 1171 | 177.61 | 1168.82 | 208.94 | 1172.12 | 455.18 | 1171.23 |
| 957 | | | 259.29 | 959.84 | 535.25 | 956.00 |
| 900 | 225.55 | 901.59 | 273.60 | 900.85 | 553.44 | 901.05 |
| 733 | | | 311.62 | 740.76 | 608.66 | 733.04 |
| 690 | 262.67 | 690.63 | 322.35 | 697.95 | 621.25 | 691.17 |
| 675 | 265.32 | 675.63 | 325.87 | 682.95 | 625.19 | 676.87 |
| 567 | 284.22 | 567.99 | 351.42 | 576.12 | 656.22 | 566.87 |
| 533 | 290.23 | 534.13 | 359.73 | 541.78 | 665.44 | 531.81 |
| 426 | | | 384.66 | 435.52 | 694.21 | 424.29 |
| 345 | | | 404.20 | 353.70 | 715.02 | 343.76 |
| 298 | | 297.41 | 415.20 | 305.71 | 726.30 | 296.24 |
| 260 | 337.46 | 259.83 | 423.76 | 266.74 | 735.03 | 259.16 |
| 231 | 342.26 | 230.74 | 430.08 | 236.92 | 741.78 | 229.47 |
| 199 | | | 437.59 | 207.09 | 749.00 | 198.57 |
| 164 | 353.83 | 164.62 | 445.64 | 170.81 | 756.66 | 163.84 |
| 137 | | | 451.87 | 143.93 | 762.76 | 136.88 |
| 25 | | | 477.72 | 31.71 | 786.37 | 25.99 |
| -3 | | | 483.78 | 5.00 | 791.91 | -1.22 |

**Table A1.** Synchronisation of the Law Dome ice core chronology to other commonly used ice core chronologies. Ant2k dates are those already published by Ahmed et al. (2013) using the mean date of the event peak in WAIS, NGRIP and DSS cores. To determine the AICC2012 date, DSS is matched to NGRIP, selected as it has the highest resolution, though fewer ties can be matched. All depths referred to in the DSS2022 chronology are from the DSSMain core except for those marked † which indicates DSS1617 and ‡ where DSS99 is used.

Overall, the results shown in Table A1 are broadly consistent across all the ice cores used in the composite record. We would not expect all of the tests to show statistical significance for several reasons: a) the accumulation rates used for the comparisons are as similar as possible, but not identical, b) even if the accumulation rates where identical we would expect variability in the species concentrations (i.e, the correlation with accumulation is not perfect) and c) the presence of surface features such as sastrugi will result in different (but highly correlated) ice core records from ice cores drilled at different locations, even only a few metres apart.

| Species | DSS1617 | | DSS99 | | DSS97 | |
|---------|---------|---------|---------|---------|---------|---------|
| | K-S | W-T | K-S | W-T | K-S | W-T |
| chloride | 0.147 | -5.032 | **0.041** | **0.553** | 0.081 | -3.737 |
| nitrate | 0.094 | -2.207 | 0.089 | -3.086 | 0.382 | -22.404 |
| sulphate | **0.080** | 2.031 | 0.143 | -4.827 | 0.097 | -5.240 |
| sodium | 0.126 | -4.428 | **0.034** | **0.273** | 0.098 | -4.718 |
| magnesium | 0.156 | -2.967 | 0.088 | 4.161 | 0.063 | **-1.722** |

**Table A1.** Statistical comparison between DSSMain and the shorter DSS97, DSS99, DSS1617 cores that make up the composite. Tests are performed between segments from cores with the closest average value of snow accumulation, as calculated using the changepoint analysis. D-statistics from the Kolmogorov-Smirnov (K-S) test and Welch's t-test (W-T) in bold indicate species where the p-value is greater than 0.05, indicating that the null hypothesis that both samples are from the same distribution cannot be rejected and hence have a comparable distribution.

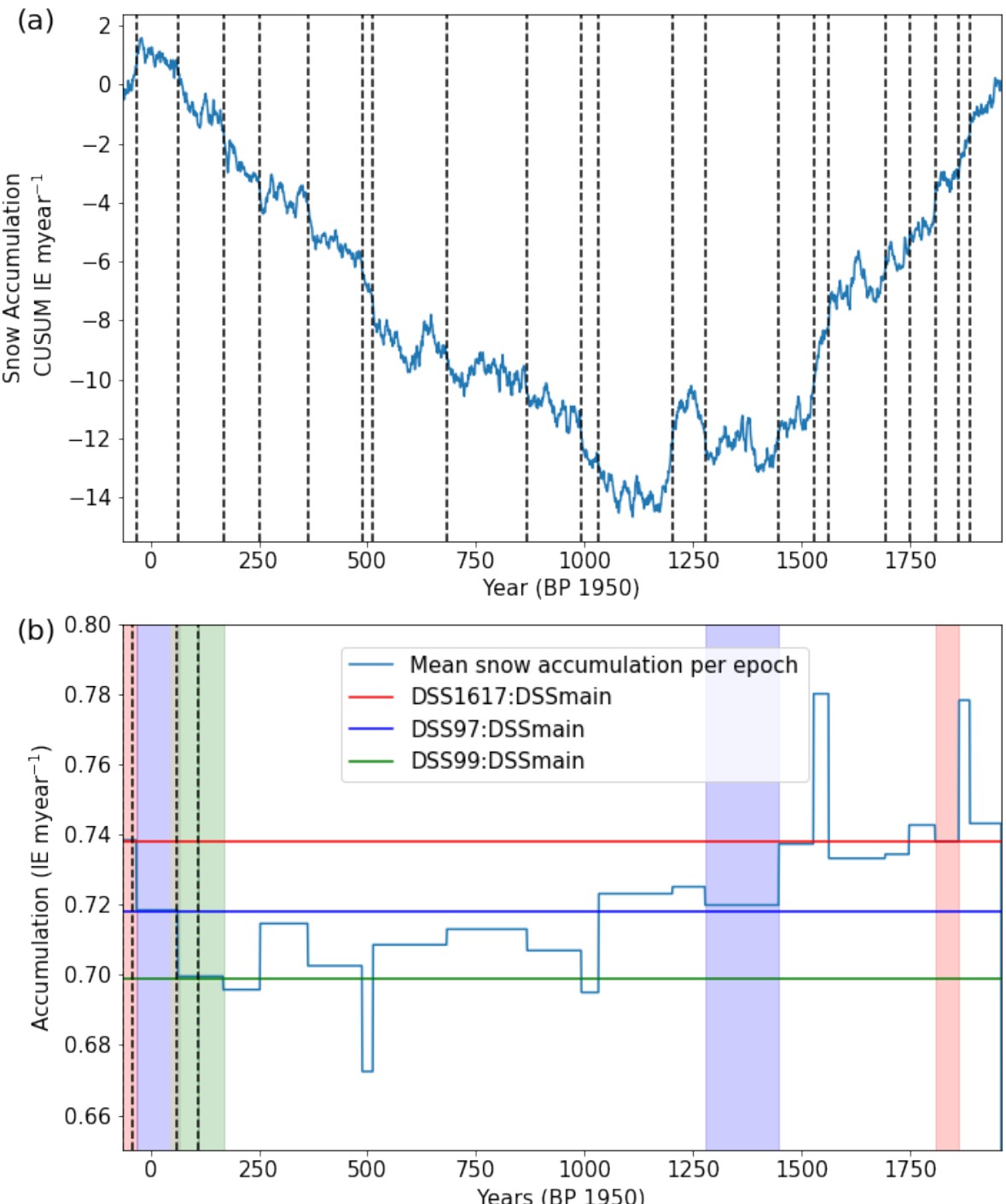

**Figure A1.** (a) Cumulative sum of the annual snow accumulation rate with detected change points indicated by vertical dashed lines. (b) Average snow accumulation epochs used for chemistry data statistical analysis.

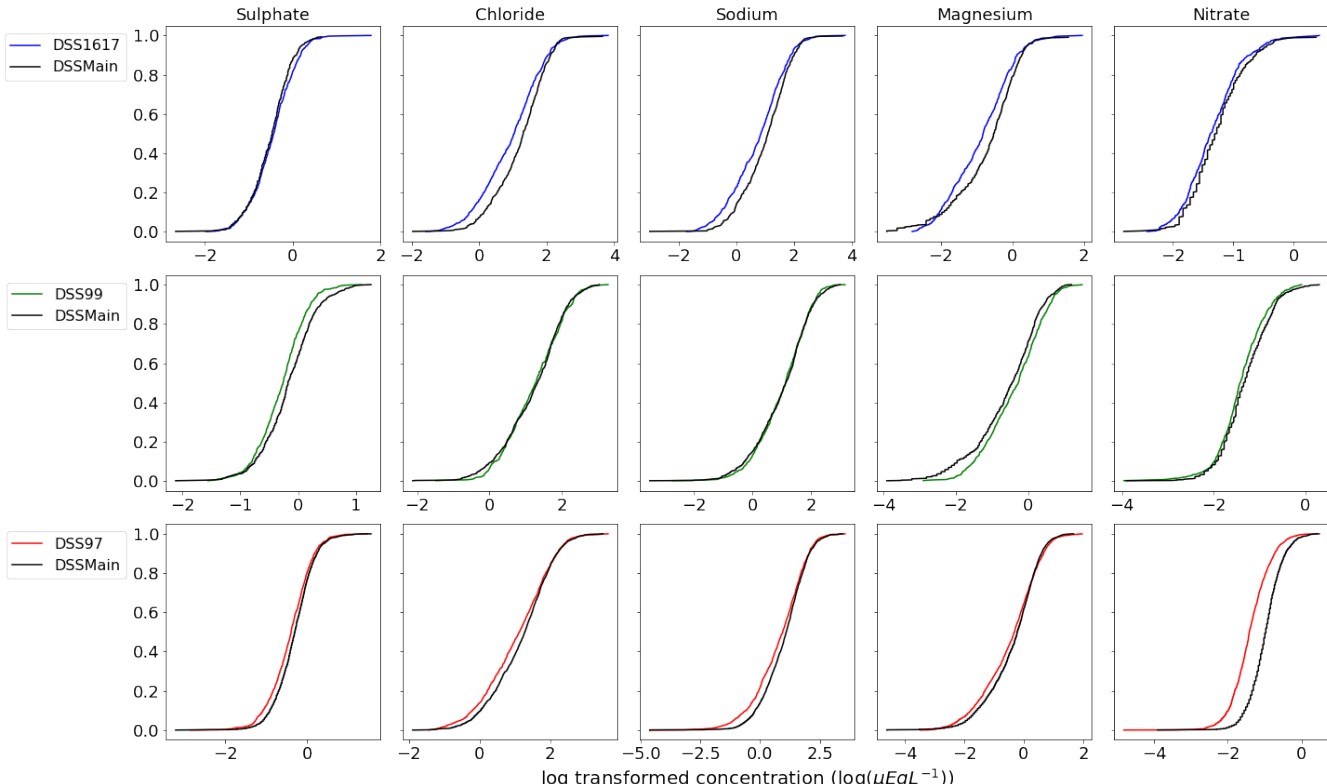

**Figure A2.** Empirical cumulative distribution functions of the log transformed concentration data for each trace chemistry species. Epochs from DSS1617, DSS99 and DSS97 are each compared against epochs from the DSSMain core with similar, but not identical, snow accumulation rates. Epochs are determined by the changepoint analysis on the CUSUM snow accumulation.

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
