# Peer review of "2000 years of annual ice core data from Law Dome, East Antarctica"

_Earth System Science Data, 2021_

## Referee Comment (RC1)

**Review ESSD paper 2021 - 408**

Dear authors. After carefully reading your article '2000 years of annual ice corer data from Law Dome, East Antarctic' I have some minor revisions that should be eliminated before publishing.

**1. Spelling mistakes and form**

Line: 1: Two times 'the'

Line 10: Delete brackets one time

Line 44: I would suggest to add the coordinates of the site.

Line 66: Space between '1' and 'm'

Line 81: be consistent: Here you write 'metre' but in the rest of the document just 'm'. If you insist to write meter here, than you should at least add a 's'.

Line 121: […] detailed results of this analysis shows → delete 's'

Line 163: '.' at the end missing

Line 165: closing bracket missing

Line 215: should be 'and' and not 'an'

Line 222: Sentence starting with 'Overall […]' does not make sense.

Table 1: I would suggest deleting .26 (DSS99) and to adjust the coordinates of DSS1617, so they have the same form as the others.

Figure 4: I would suggest adding information about the cores here. It would make it much easier to understand.

Line 130: Add reference

General:

Make sure you write either the number or the word (concerning number until 10). E. g. in line 82 you write '7', but in line 9 'four' decades.

Make sure you write either 'level' or 'Level'.

Make sure you write 'Fig.' in the text and not 'Figure' (at least be consistent).

Make sure you write either DSSMain or DSSmain

Paragraph 'Trace Ion Chemistry' → I suggest including an overview / table with the different methods used for the different cores as well as the reference.

Appendix A: Why are you suddenly writing about 'core A' and 'core B'

**Content**

In general I am missing in the paper the uniqueness and benefit for further research and other scientists of this dataset. There is neither a sentence about this in the abstract nor in the summary. As one of the main goals of ESSD products is the utility of the data ('authors should know, […] the data product interest a sufficient number of users'.)

→ Please describe in much more detail, who benefits from the dataset, which kind of researchers might be interested in the products, and why is the new dataset much more useful than the old one.

Also be sure you make the validity and applicability of your dataset more clearly within the paper.

**Dataset**

Add information about 'coring devices'

Add information about missing or insufficient data (you write about this in e. g.  Appendix A in the paper)

Add information (see above) in the metadata about the purpose. 'Why was/ is this study so important' (benefits for other researchers).

---

## Community Comment (CC1)

**Comment on ice-core chronology, volcanic eruptions and solar proton events in 774 and 993**

Thanks for the authors for describing and making available this data from an exceptionally high-resolved ice-core record from East Antarctica. I would here like to add some perspective on chronological questions, which I believe would be helpful to address to raise awareness in the broader research community and potential users of the data.

The DSS layer counted timescale was derived independently without reference to volcanic eruptions or other potential stratigraphic age markers. This approach is thus very useful in quantifying uncertainties of annual-layer counting in ice cores, which cannot be done if external age constraints were used during development of the dating. You provide subjective quantitate estimates of the dating uncertainty based on the number and quality of independent seasonal tracers and consider your age error estimates as "conservative" and rather biased towards undercounting than over-counting.

I wonder what is your basis for these considerations? Can these be underpinned by existing data or analyses? Undercounting would be the exemption for ice-core chronologies based on annual-layer counting. For the Holocene and Common Era (see Table 1), at least three different annual-layer counted chronologies exist: (1) *Meese/Sowers* for GISP2 (Meese 1999), (2) *GICC05* for Dye3, GRIP and NGRIP1 (all in Greenland (Vinther et al., 2006)), and (3) *WDC06A-7* for WDC (Antarctica (Sigl et al., 2013)). All three of these chronologies were subject to over-counting (up to 70-80 years at 11 ka BP as was established with the help of cosmogenic radionuclides (i.e. $^{14}$C, $^{10}$Be; (Muscheler et al., 2014; Sigl et al., 2016)). Following the discovery of global-scale anomalies in the tree-ring $^{14}$C content in trees (Büntgen et al., 2018; Miyake et al., 2012) and subsequent detection of corresponding anomalies (up to +9 standard deviations from natural variability) in $^{10}$Be ice-core concentrations (Miyake et al., 2015; Sigl et al., 2015), a dating bias (towards c. 7 years too old at 775 CE) was established in Greenland (2) and Antarctica (3) chronologies. Details about the reasons explaining this bias are presented elsewhere (e.g. Plunkett et al., 2022; Sinnl et al. 2021). Consequently, new annual-layer counted chronologies were constructed constrained by the 775 CE anomaly (see Figure 1). In Greenland this was (4) *NS1-2011* for NEEM(2011-S1), and (5) *DRI_NGRIP2* for NGRIP2 (5) and in Antarctica this was (6) *WD2014* for WDC (replacing WDC06A-7).

[Figure]

**Figure 1:** High-resolution $^{10}$Be data from NEEM(2011S1) on the NS1-2011 chronology and WDC on the WD2014 chronology (Sigl et al., 2015) relative to $\Delta^{14}$C derived from a Northern Hemisphere (NH) tree-ring stack (Büntgen et al., 2018). Shading indicates the error estimate for the solar proton event in 774 CE.

**Table 1:** Selection of key annual-layer counted ice-core chronologies.

| ID | Chronology | Region | Ice Core(s) | Age Constraints | References |
|------|-------------|------------|-------------------|--------------------------|-------------------------|
| (1) | Meese/Sowers | Greenland | GISP2 | unknown | Meese (1999) |
| (2) | GICC05 | Greenland | Dye3, GRIP, NGRIP1 | Volcanoes (V) | Vinther et al., (2006) |
| (3) | WDC06A-7 | Antarctica | WDC | V | Sigl et al. (2013) |
| (4) | NS1-2011 | Greenland | NEEM(2011-S1) | V, $^{10}$Be anomaly | Sigl et al. (2015) |
| (5) | DRI_NGRIP2 | Greenland | NGRIP2 | V, $^{10}$Be anomaly | McConnell et al. (2018) |
| (6) | WD2014 | Antarctica | WDC | V, $^{10}$Be anomaly | Sigl et al. (2015) |
| (7) | ANT2k | Antarctica | DSS, WDC, NGRIP1 | V | PAGES2k Cons. (2013) |
| (8) | DSS | Antarctica | DSS | none | Jong et al. (2022) |

The revised and constrained chronologies (4,6) have been used to reconstruct volcanic aerosol forcing recommended by PMIP4/CMIP for *past2k* climate simulations (Jungclaus et al., 2017; Toohey and Sigl, 2017). It could further be demonstrated that only by using the volcanic forcing on these chronologies consistent temperature responses could be detected in absolute-dated tree-ring temperature reconstructions throughout the last 2000-2500 years (Büntgen et al., 2020; Sigl et al., 2015).

In parallel to these chronological developments within the ice-core community, multi-proxy reconstructions and data assimilations of regional (including Antarctica) and later global temperature and precipitation have been compiled (PAGES2k Consortium 2013, 2017, Konecky et al., 2020; Steiger et al., 2018; Stenni et al., 2017). Instead of using the independent chronologies for this purpose a common chronology, named *ANT2k* (7) has been produced for Antarctica by averaging the ages of *GICC05*, *WDC06A-7* and DSS (8) for c.40 common volcanic marker events (PAGES 2k Consortium 2013). To my best knowledge (and with the exception of WDC for which the data was later updated to *WD2014*), this common chronology is used for all subsequent composite reconstructions from Antarctica within the PAGES2k Consortium. The previous mentioned revisions made on two out of the three chronologies underpinning ANT2k (largely consistent with an age shift of 5-10 years towards younger ages during the 1st millennium CE) have not yet been implemented in these major paleoclimate databases. This means that volcanic forcing and a large body of paleoclimate evidence remain temporally offset throughout most of the first millennium. Implications of this mismatch are exemplified by studies focusing on the Northern Hemisphere (Büntgen et al., 2020; Plunkett et al., 2022; Sigl et al., 2015) with far-reaching consequences for our limited understanding of amplitudes of past natural climate variability (PAGES 2k Consortium 2019; Neukom et al., 2019) culminating for example in strongly contested (Anchukaitis et al., 2012; Büntgen et al., 2018) claims of missing tree rings and alleged chronological errors in tree-ring reconstructions that accumulate back in time (Mann, 2021; Mann et al., 2012).

I am describing these lengthy details on the history of ice-core chronologies in the last decades not at all to convince the authors to change the chronology or the datasets presented in this paper. A fully independent, year-by-year history of aerosols, snow-accumulation and stable-isotopes is a very valuable asset in the research field, as demonstrated recently (Vance et al., 2022). However, there will also continue to be research applications (e.g. analyzing spatial response to rapid climate changes (Buizert et al., 2018) or estimating mean volcanic sulfate deposition across Antarctica (Sigl et al., 2014)) in which a common chronology over Antarctica will be required. In this case, it would be mandatory to know existing age differences among ice-cores.

Investigating the age differences between DSS and *WD2014* for some outstandingly large common volcanic eruption signals it appears that DSS is consistently older than WD2014 during the 1st millennium CE sometimes slightly outside the stated error bounds which you consider conservative for DSS (Table 2; Figure 2). Owing to the absence of major global-scale volcanic eruption signals between c.680 and 1230 any additional and absolute dated age marker would be very helpful to explore this age offset further. The 774 and 993 CE solar proton events previously identified in Antarctica are in my view ideal to overcome this lack of information, and to objectively evaluate the stated uncertainty bounds for DSS.

[Figure]

**Figure 2:** Volcanic matching between WDC and DSS; **a:** age difference between WD2014 and DSS for common major volcanic marker events. Positive values indicate younger ages for WDC during the 1st millennium; **b:** measured or inferred (for WDC sulfur was analyzed) sulfate concentrations between 400-700 CE with the ages of the used matching-points (see Table 2 for details) indicated using the WD2014 timescale (blue) and DSS (black). Note these do not indicate the eruption dates.

**Table 2:** Volcanic synchronization between annual-layer counted ice-core chronologies DSS and WD2014.

| Year WD2014 (CE) | Year DSS (CE) | age difference (WD2014 minus DSS) years | WD2014 error (+/-) years | DSS Depth (m) | WD Depth (m) |
|---|---|---|---|---|---|
| 1816.0 | 1816.3 | -0.3 | 0 | 133.41 | 59.03 |
| 1810.4 | 1810.5 | -0.1 | 0 | 137.04 | 60.49 |
| 1695.9 | 1695.8 | 0.1 | 1 | 204.04 | 87.82 |
| 1642.5 | 1642.5 | -0.1 | 0 | 234.48 | 100.47 |
| 1459.7 | 1459.6 | 0.1 | 2 | 328.29 | 142.96 |
| 1277.3 | 1276.7 | 0.6 | 1 | 411.39 | 184.59 |
| 1269.6 | 1268.7 | 0.9 | 1 | 414.72 | 186.23 |
| 1258.8 | 1258.3 | 0.6 | 1 | 418.89 | 188.71 |
| 1241.7 | 1241.2 | 0.5 | 1 | 426.00 | 192.82 |
| 1230.5 | 1229.8 | 0.7 | 2 | 430.68 | 195.38 |
| 682.9 | 677.1 | 5.9 | 2 | 625.14 | 325.87 |
| 576.0 | 566.9 | 9.1 | 2 | 656.21 | 351.44 |
| 541.8 | 531.6 | 10.2 | 2 | 665.50 | 359.73 |
| 435.7 | 424.1 | 11.6 | 5 | 694.27 | 384.63 |
| 170.8 | 163.9 | 6.9 | 5 | 756.64 | 445.63 |

To summarize, I suggest the authors should consider:

1) To weaken the language that the DSS layer-counting errors are conservative.
2) To analyze in high-resolution [10]Be around 774 and 993 CE for evaluation of the chronology.
3) To include a table as supporting information providing the ice-core depths for the previously identified common age markers in NGRIP1, WDC and DSS allowing age-transfer between ice cores.
4) Provide some information in the main text that DSS is independent but not synchronous with the *WD2014* chronology commonly used for synchronizing ice cores in Antarctica – and a backbone for volcanic forcing reconstructions. For ongoing initiatives such as PAGES CLIVASH2k this information will be helpful, since it can help to avoid the smoothing and amplitude loss inherent in the stacking of proxy records with limited age synchronization (e.g. PAGES 2k Consortium 2019) which remains a major limitation for our understanding of natural climate variations.

**References:**

Anchukaitis, K. J., Breitenmoser, P., Briffa, K. R., Buchwal, A., Buntgen, U., Cook, E. R., D'Arrigo, R. D., Esper, J., Evans, M. N., Frank, D., Grudd, H., Gunnarson, B. E., Hughes, M. K., Kirdyanov, A. V., Korner, C., Krusic, P. J., Luckman, B., Melvin, T. M., Salzer, M. W., Shashkin, A. V., Timmreck, C., Vaganov, E. A., and Wilson, R. J. S.: Tree rings and volcanic cooling, *Nat Geosci*, 5, 836-837, 2012.

Buizert, C., Sigl, M., Severi, M., Markle, B. R., Wettstein, J. J., McConnell, J. R., Pedro, J. B., Sodemann, H., Goto-Azuma, K., Kawamura, K., Fujita, S., Motoyama, H., Hirabayashi, M., Uemura, R., Stenni, B., Parrenin, F., He, F., Fudge, T. J., and Steig, E.: Abrupt Ice Age Shifts in Southern Westerlies and Antarctic Climate Forced from the North, *Nature*, 2018. 543-549, 2018.

Büntgen, U., Arseneault, D., Boucher, É., Churakova, O. V., Gennaretti, F., Crivellaro, A., Hughes, M. K., Kirdyanov, A. V., Klippel, L., Krusic, P. J., Linderholm, H. W., Ljungqvist, F. C., Ludescher, J., McCormick, M., Myglan, V. S., Nicolussi, K., Piermattei, A., Oppenheimer, C., Reinig, F., Sigl, M., Vaganov, E. A., and Esper, J.: Prominent role of volcanism in Common Era climate variability and human history, *Dendrochronologia*, 64, 125757, 2020.

Büntgen, U., Wacker, L., Galván, J. D., Arnold, S., Arseneault, D., Baillie, M., Beer, J., Bernabei, M., Bleicher, N., Boswijk, G., Bräuning, A., Carrer, M., Ljungqvist, F. C., Cherubini, P., Christl, M., Christie, D. A., Clark, P. W., Cook, E. R., D'Arrigo, R., Davi, N., Eggertsson, Ó., Esper, J., Fowler, A. M., Gedalof, Z. e., Gennaretti, F., Grießinger, J., Grissino-Mayer, H., Grudd, H., Gunnarson, B. E., Hantemirov, R., Herzig, F., Hessl, A., Heussner, K.-U., Jull, A. J. T., Kukarskih, V., Kirdyanov, A., Kolář, T., Krusic, P. J., Kyncl, T., Lara, A., LeQuesne, C., Linderholm, H. W., Loader, N. J., Luckman, B., Miyake, F., Myglan, V. S., Nicolussi, K., Oppenheimer, C., Palmer, J., Panyushkina, I., Pederson, N., Rybníček, M., Schweingruber, F. H., Seim, A., Sigl, M., Churakova, O., Speer, J. H., Synal, H.-A., Tegel, W., Treydte, K., Villalba, R., Wiles, G., Wilson, R., Winship, L. J., Wunder, J., Yang, B., and Young, G. H. F.: Tree rings reveal globally coherent signature of cosmogenic radiocarbon events in 774 and 993 CE, *Nat Commun*, 9, 3605, 2018.

Jungclaus, J. H., Bard, E., Baroni, M., Braconnot, P., Cao, J., Chini, L. P., Egorova, T., Evans, M., Gonzalez-Rouco, J. F., Goosse, H., Hurtt, G. C., Joos, F., Kaplan, J. O., Khodri, M., Goldewijk, K. K., Krivova, N., LeGrande, A. N., Lorenz, S. J., Luterbacher, J., Man, W. M., Maycock, A. C., Meinshausen, M., Moberg, A., Muscheler, R., Nehrbass-Ahles, C., Otto-Bliesner, B. I., Phipps, S. J., Pongratz, J., Rozanov, E., Schmidt, G. A., Schmidt, H., Schmutz, W., Schurer, A., Shapiro, A. I., Sigl, M., Smerdon, J. E., Solanki, S. K., Timmreck, C., Toohey, M., Usoskin, I. G., Wagner, S., Wu, C. J., Yeo, K. L., Zanchettin, D., Zhang, Q., and Zorita, E.: The PMIP4 contribution to CMIP6-Part 3: The last millennium, scientific objective, and experimental design for the PMIP4 past1000 simulations, *Geosci Model Dev*, 10, 4005-4033, 2017.

Konecky, B. L., McKay, N. P., Churakova, O. V., Comas-Bru, L., Dassie, E. P., DeLong, K. L., Falster, G. M., Fischer, M. J., Jones, M. D., Jonkers, L., Kaufman, D. S., Leduc, G., Managave, S. R., Martrat, B., Opel, T., Orsi, A. J., Partin, J. W., Sayani, H. R., Thomas, E. K., Thompson, D. M., Tyler, J. J., Abram, N. J., Atwood, A. R., Cartapanis, O., Conroy, J. L., Curran, M. A., Dee, S. G., Deininger, M., Divine, D. V., Kern, Z., Porter, T. J., Stevenson, S. L., von Gunten, L., and Iso2k, P.: The Iso2k database: a global compilation of paleo-delta O-18 and delta H-2 records to aid understanding of Common Era climate, *Earth System Science Data*, 12, 2261-2288, 2020.

Mann, M. E.: Beyond the hockey stick: Climate lessons from the Common Era, *P Natl Acad Sci USA*, 118, 2021.

Mann, M. E., Fuentes, J. D., and Rutherford, S.: Underestimation of volcanic cooling in tree-ring-based reconstructions of hemispheric temperatures, *Nat Geosci*, 5, 202-205, 2012.

Miyake, F., Nagaya, K., Masuda, K., and Nakamura, T.: A signature of cosmic-ray increase in AD 774-775 from tree rings in Japan, *Nature*, 486, 240-242, 2012.

Miyake, F., Suzuki, A., Masuda, K., Horiuchi, K., Motoyama, H., Matsuzaki, H., Motizuki, Y., Takahashi, K., and Nakai, Y.: Cosmic ray event of AD 774-775 shown in quasi-annual 10 Be data from the Antarctic Dome Fuji ice core, *Geophys. Res. Lett.*, 2015.

Muscheler, R., Adolphi, F., and Knudsen, M. F.: Assessing the differences between the IntCal and Greenland ice-core time scales for the last 14,000 years via the common cosmogenic radionuclide variations, *Quaternary Sci Rev*, 106, 81-87, 2014.

Neukom, R., Steiger, N., Gomez-Navarro, J. J., Wang, J. H., and Werner, J. P.: No evidence for globally coherent warm and cold periods over the preindustrial Common Era, *Nature*, 571, 550-554, 2019.

PAGES 2k Consortium: Continental-scale temperature variability during the past two millennia, *Nat Geosci*, 6, 339-346, 2013.

PAGES2k Consortium: Data Descriptor: A global multiproxy database for temperature reconstructions of the Common Era, *Sci Data*, 4, 2017.

PAGES 2k Consortium: Consistent multidecadal variability in global temperature reconstructions and simulations over the Common Era, *Nat Geosci*, 12, 643-649, 2019.

Plunkett, G., Sigl, M., Schwaiger, H. F., Tomlinson, E. L., Toohey, M., McConnell, J. R., Pilcher, J. R., Hasegawa, T., and Siebe, C.: No evidence for tephra in Greenland from the historic eruption of Vesuvius in 79 CE: implications for geochronology and paleoclimatology, *Clim. Past*, 18, 45-65, 2022.

Sigl, M., Fudge, T. J., Winstrup, M., Cole-Dai, J., Ferris, D., McConnell, J. R., Taylor, K. C., Welten, K. C., Woodruff, T. E., Adolphi, F., Bisiaux, M., Brook, E. J., Buizert, C., Caffee, M. W., Dunbar, N. W., Edwards, R., Geng, L., Iverson, N., Koffman, B., Layman, L., Maselli, O. J., McGwire, K., Muscheler, R., Nishiizumi, K., Pasteris, D. R., Rhodes, R. H., and Sowers, T. A.: The WAIS Divide deep ice core WD2014 chronology - Part 2: Annual-layer counting (0-31 ka BP), *Clim Past*, 12, 769-786, 2016.

Sigl, M., McConnell, J. R., Layman, L., Maselli, O., McGwire, K., Pasteris, D., Dahl-Jensen, D., Steffensen, J. P., Vinther, B., Edwards, R., Mulvaney, R., and Kipfstuhl, S.: A new bipolar ice core record of volcanism from WAIS Divide and NEEM and implications for climate forcing of the last 2000 years, *J Geophys Res-Atmos*, 118, 1151-1169, 2013.

Sigl, M., McConnell, J. R., Toohey, M., Curran, M., Das, S. B., Edwards, R., Isaksson, E., Kawamura, K., Kipfstuhl, S., Kruger, K., Layman, L., Maselli, O. J., Motizuki, Y., Motoyama, H., Pasteris, D. R., and Severi, M.: Insights from Antarctica on volcanic forcing during the Common Era, *Nat Clim Change*, 4, 693-697, 2014.

Sigl, M., Winstrup, M., McConnell, J. R., Welten, K. C., Plunkett, G., Ludlow, F., Büntgen, U., Caffee, M., Chellman, N., Dahl-Jensen, D., Fischer, H., Kipfstuhl, S., Kostick, C., Maselli, O. J., Mekhaldi, F., Mulvaney, R., Muscheler, R., Pasteris, D. R., Pilcher, J. R., Salzer, M., Schupbach, S., Steffensen, J. P., Vinther, B. M., and Woodruff, T. E.: Timing and climate forcing of volcanic eruptions for the past 2,500 years, *Nature*, 523, 543-549, 2015.

Sinnl, G., Winstrup, M., Erhardt, T., Cook, E., Jensen, C., Svensson, A., Vinther, B. M., Muscheler, R., and Rasmussen, S. O.: A multi-ice-core, annual-layer-counted Greenland ice-core chronology for the last 3800 years: GICC21, *Clim. Past Discuss.* [preprint], https://doi.org/10.5194/cp-2021-155, in review, 2021.

Steiger, N. J., Smerdon, J. E., Cook, E. R., and Cook, B. I.: Data Descriptor: A reconstruction of global hydroclimate and dynamical variables over the Common Era, *Sci Data*, 5, 2018.

Stenni, B., Curran, M. A. J., Abram, N. J., Orsi, A., Goursaud, S., Masson-Delmotte, V., Neukom, R., Goosse, H., Divine, D., Van Ommen, T., Steig, E. J., Dixon, D. A., Thomas, E. R., Bertler, N. A. N., Isaksson, E., Ekaykin, A., Werner, M., and Frezzotti, M.: Antarctic climate variability on regional and continental scales over the last 2000 years, *Clim Past*, 13, 1609-1634, 2017.

Toohey, M. and Sigl, M.: Volcanic stratospheric sulfur injections and aerosol optical depth from 500 BCE to 1900 CE, *Earth System Science Data*, 9, 809-831, 2017.

Vance, T. R., Kiem, A. S., Jong, L. M., Roberts, J. L., Plummer, C. T., Moy, A. D., Curran, M. A. J., and van Ommen, T. D.: Pacific decadal variability over the last 2000 years and implications for climatic risk, *Communications Earth & Environment*, 3, 33, 2022.

Vinther, B. M., Clausen, H. B., Johnsen, S. J., Rasmussen, S. O., Andersen, K. K., Buchardt, S. L., Dahl-Jensen, D., Seierstad, I. K., Siggaard-Andersen, M. L., Steffensen, J. P., Svensson, A., Olsen, J., and Heinemeier, J.: A synchronized dating of three Greenland ice cores throughout the Holocene, *J Geophys Res-Atmos*, 111, 2006.

---

## Author Comment (AC1)

**2000 years of annual ice core data from Law Dome, East Antarctica Response to reviewer comments**

We thank the reviewers for their positive response to our paper and the helpful feedback provided in their reviews which will help improve the quality of the manuscript. We have addressed all comments from the three reviews as well as the community comment in this one response document.

There were several comments made by multiple reviewers that were very similar, we will address these first and then respond to the more detailed comments of the individual reviewers.

All three reviewers commented about the confusing structure of the paper with text and figures appearing out of order and questioning the need for Appendices. We agree that the current placement of many figures and tables is not optimal and we have adjusted the layout to ensure they appear closer to the text in which they are referred to. In addition we have removed the figures and tables that were previously in Appendix B, incorporating the data file headers and plots of the individual time series into main text of the paper. The description of the extra statistical analysis remains in Appendix A, which will move into a separate supplementary information document.

All three reviewers commented on the need for more explanation about why the dataset is useful, including outside of just the ice core and paleoclimate communities. We included and extended discussion about the usefulness and validity of the datasets. We agree that more detail on this in the paper is useful, and we have included this information in the abstract and have made several changes to the introduction, particularly the paragraph starting at line 36 in the revised manuscript. Please see the tracked changes document to see the full details of the text added.

All three reviewers made very useful detailed comments including some typos. We address these now specifically referring to each reviewer.

**Reviewer 1** Line: 1: Two times 'the' We have removed the extra 'the'.

Line 10: Delete brackets one time

Bracket is deleted, this now reads: over the past 2000 years e.g. Stenni et al., 2017)

Line 44: I would suggest to add the coordinates of the site.

We have added the the co-ordinates. The sentence now reads: The DSS site is located at 66°46'11"S 112°48'25"E, approximately 4.7 km SSW from the dome summit (Morgan et al., 1997).

Line 66: Space between '1' and 'm'

We have added a space and corrected all other instances where a space is needed between the number and unit.

Line 81: be consistent: Here you write 'metre' but in the rest of the document just 'm'. If you insist to write meter here, than you should at least add a 's'.

We have changed this to read '30 m' and use the abbreviated units throughout the

manuscript.

Line 121: [...] detailed results of this analysis shows  $\rightarrow$  delete 's' We have deleted the extra 's'

Line 163: '.' at the end missing We have added '.' at the end of the sentence.

Line 165: closing bracket missing We have added the closing bracket.

Line 215: should be 'and' and not 'an' This has been corrected to 'and'.

Line 222: Sentence starting with 'Overall [...]' does not make sense. We have edited this sentence for clarity as follows:

Overall, the results shown in Table A1 show the results are broadly consistent across the various ice cores all the ice cores used in the composite record.

Table 1: I would suggest deleting .26 (DSS99) and to adjust the coordinates of DSS1617, so they have the same form as the others.

We have deleted the extra decimal places and adjusted the coordinates of DSS1617 to have the same form as the others.

Figure 4: I would suggest adding information about the cores here. It would make it much easier to understand.

We are unsure what is meant by this comment, the histograms in Figure 4 are of the distributions of the concentrations of all trace chemistry analytes of each sample used in the composite record. Perhaps the reviewer meant Fig 3? We have added a reference to Table 1 in the figure caption to make this clearer.

Line 130: Add reference

We have added text to this sentence for extra clarity. It now reads: The density observations and empirical fit are shown in Figure 5. as a function of depth (see Eqn. 1) are plotted in Fig. 5, showing good agreement between the two.

Make sure you write either the number or the word (concerning number until 10). E. g. in line 82 you write '7', but in line 9 'four' decades.

We have edited the manuscript and now only write the number, not the word.

Make sure you write either 'level' or 'Level'.

We have altered the manuscript to only use 'level', unless capitalisation is required at the start of a sentence.

Make sure you write 'Fig.' in the text and not 'Figure' (at least be consistent). We have altered the manuscript to use 'Fig.' throughout.

Make sure you write either DSSMain or DSSmain We have ensured that we now use DSSMain throughout the manuscript. Paragraph 'Trace Ion Chemistry' I suggest including an overview / table with the different methods used for the different cores as well as the reference.

We have added a table of the different methods used for the cores and included a reference to supplement the text about the different methods used.

Appendix A: Why are you suddenly writing about 'core A' and 'core B'

We agree that this terminology could be confusing and had meant to simply refer to any two arbitrary cores. We have now modified this text as follows:

Therefore, as an alternative, we identify different epochs when we might expect similar distributions, specifically periods of time in ice core A covered by one of the ice cores where the average accumulation rate approximately the same as ice core B in a different core making up the compilation but during a different period of time.

In general I am missing in the paper the uniqueness and benefit for further research and other scientists of this dataset. There is neither a sentence about this in the abstract nor in the summary. As one of the main goals of ESSD products is the utility of the data ('authors should know, [...] the data product interest a sufficient number of users'.)  $\rightarrow$  Please describe in much more detail, who benefits from the dataset, which kind of researchers might be interested in the products, and why is the new dataset much more useful than the old one.

Please see our response above about the additions we have made to the manuscript to include this information.

Also be sure you make the validity and applicability of your dataset more clearly within the paper.

We have included in the summary of the manuscript a list to summarise any caveats and where the dataset is applicable.

Add information about 'coring devices'

We will add further information in the dataset metadata about the coring devices used to obtain the 4 different cores in this composite.

Add information about missing or insufficient data (you write about this in e. g. Appendix A in the paper)

We have added information in this in the metadata for the dataset.

Add information (see above) in the metadata about the purpose. 'Why was/ is this study so important' (benefits for other researchers).

We have added more information to the metadata record for the dataset using similar text as is referred to above.

**Reviewer 2** This data description paper by Lenneke Jong and co-authors presents composite records of oxygen isotopic composition, chemistry and accumulation rate from the Law Dome site in East Antarctica. Both raw data (against depth) and mean annual values are provided except for the isotopic record, where only level2 data are presented. While many citations are in the text about the use of these data, the authors should improve a bit on why these data are useful and to which communities apart from ice core and paleoclimate ones, for which it is clear. Please see our responses above, we have now added this information to the manuscript.

The paper accompanying the data is well written, but I would suggest improving the structure. Sometimes I found some difficulties in understanding the differences between Main text, Appendices, Supplementary material with some figures inside the text and others at the end. It confuses ...

We agree that the structure of the manuscript was at times confusing. Please see the response to the similar comment above, we believe we have rearranged the figures and text to reduce confusion.

Moreover, when trying to access the data, clicking on the link at the end of the abstract and also in the Data Availability Section, a prompt asked me for an email address before continuing. On the other end, when I entered the section "View the data set contents" I was able to download the data. Please fix this.

Unfortunately this is a feature of the AADC to prevent excessive automated downloads and we are unable to disable it. We can add instructions to assist users of the data.

Lines 56-57: "The DSS record currently spans -11 to 2017 CE..." If I look in the online description, a -9 is reported. Please, fix.

Thank you for pointing out this inconsistency. The record spaces -11 to 2017 CE and we have amended the manuscript and the metadata entry at the AADC to make this consistent this throughout.

Line 75: the length of the core DSS main and the drilling period are different in the text and in the Table 1.

Thank you for pointing this out, the lengths in Table 1 have been amended to be consistent with the correct values in the text.

Lines 89-90: to be honest it is not clear to me if the level 1 isotopic data will change in the future and so will change also the mean annual values of this composite ... .. Am I wrong?

The mean annual values for the isotopic data will not change as the underlying isotopic data will not change. This data is complete for these cores, but has not yet been released as a manuscript is in preparation which is to be the preferred initial publication for this data.

In the trace Ion Chemistry section 3.1 I do find information on the precisions of the different analyses published in different papers .... I would recommend some reorganization (adding a table perhaps).

We have added a table (Table 3) in the updated manuscript to summarise this information consisely.

The figure 4 and Table 2 are never cited in the text ... please add.

We have added references to both figure 4 and table 2 in the text, noting that Table 2 is now Table 4 due to the reorganisation on the manuscript.

At line 152 we add:

Summary statistics for the level 1 trace chemistry species are included in Table 4, with the corresponding histograms shown for each species shown in Fig. 4.

Line 142: figure 6 is at the end of the paper .... Why?

We realise that the default layout produced by LaTeX has resulted in a more confusing manuscript. We have adjusted this to make sure Figure 6, and the other figures have been moved to more appropriate places in the manuscript.

Line 146: see my comments above (from -9 or from -11??). Please refer to our response above.

Line 163: Figure 7 at the end of the paper. Why?

Please refer to our responses above about the structure of the manuscript and figure 6 specifically

Line 190: please explain the negative concentration values in figures B9 to B11.

Thank you for bringing this potentially confusing detail to our attention. The negative values is the result of the log transformation performed on the concentration values of chemistry data. We have added a sentence to make this clearer.

In Appendix B, I found very useful the explanation of the data file headers but some of them are after the Reference section ... again I would suggest some reorganizations of the structure.

We have reorganised the structure of the paper to incorporate these data file headers into the sections for each of the data streams, which we believe should make the manuscript easier to follow.

From figure B1 to B6, please, add in the figure captions the explanation of the panel at the bottom.

We have added in each of the captions the following sentence "The lower panel indicates the number of individual samples in the year used for the average value." Please note that each of these figures is now contained in the main text of the manuscript in their appropriate sections.

**Reviewer 3 Elizabeth Thomas**

The paper presents the ice core data from Law Dome, extending the previously published works and updating some of the available datasets. It sets a great example to the community to make the data from this iconic site available. The paper is well-written, provides the appropriate level of detail and I recommend it is accepted for publication. I only have a few minor suggestions for the text and figures below. While I appreciate the importance of updating previously published records, I do agree that the paper might be improved if you specified why this is the case. Speaking as a user of ice core datasets I appreciate the good practice. It is also refreshing to see an age-scale revisited and more generous errors applied. However, perhaps you could spell this out in the introduction as justification for this work. For example, for large-scale reconstructions, data compilations or data-model comparisons (e.g., the PAGES compilations), it is essential that the most accurate records are available and citable.

We thank Elizabeth Thomas for this encouraging review and for recognising the importance of the Law Dome dataset. We acknowledge that we should add more details as to why this dataset is important and which researchers, particularly from outside of the ice core and palaeoclimate communities will find it useful. Please see above in our responses to the common reviewer comments where we have addressed this.

I found the jump between main paper, figures, and appendix a bit difficult to follow. For this style of data paper is an appendix needed? Or could the figures be included in the main text?

We have attempted to better organise the figures and tables into more logical positions within the main manuscript. We felt that the plots of the individual time series could be left to the end with the main combined plots left within the text itself.

Line 50 – I found the word sequence "seasonal species variations" a bit odd. Perhaps "seasonal deposition of species"?

We agree this is a confusing phrase, this sentence now reads: "The DSS record has been dated using seasonal species variations variations in the deposition of species to define calendar year boundaries, commonly known as annual layer counting.

Table 2. It might be useful to include the time in the caption. Are these values calculated for the full length of the record?

These statistics are calculated over the 2000 years annually dated ice core records described in this paper. Note that after reorganising the structure of the paper, this is now numbered as Table 4. We have included a sentence in the caption: Summary statistics for the level 1 data for each analysed species, calculated over the for the full 2000 years of data included in this compilation.

Line 165 – closed bracket needed. We have added the closing bracket.

Line 171 – consider rephrasing. This sentence suggests that there are two definitions for summer which doesn't read well. What you mean is that different seasonal separations have been applied in different studies. But on its own, this sentence seems to suggest there is a summer and a warm period.

We have rephrased the section to describe the three different seasonal separations more definitively as an aggregation of the level one sea salt data into three separate (log-transformed) seasonal means (DJFM, DJFMAM and JJASON). We then describe the purposes they have so far been applied to in terms of Pacific decadal climate reconstruction and regional climate proxies.

**4.4. This data was also used in the PAGES 2k data compilation for snow accumulation. So, I assume this also needs to be updated?**

We thank the reviewer for pointing this out. This data is indeed an update from the data that was included in the PAGES 2k snow accumulation compilation and any updates to that should include this new data. We have added the following text to draw this to the readers attention: This record has been previously published in Roberts et al. (2015) and and included in PAGES 2k snow accumulation data compilation (Thomas et al., 2017). This update includes the updated here to include the newer DSS1617 core data and the improved dating.

Figure 6 – Perhaps align the legends? Cl and snow accumulation headings seem out of

place.

Thank you, we have altered this figure to align the legends and made them more transparent of the legend so as not to cover any of the data.

Figure A2 – Please increase the font site for the titles and legend. Agreed, we have updated the figure in new version of the manuscript uses increased font size for this image.

**Community comment by Sigl** The community comment posted by Michael Sigl asks several questions about the dating, uncertainties in the dating and how it compares to other ice core chronologies. We would like to thank him for taking the time to produce such a thorough and thoughtful response to our discussion paper. We agree that there is much value in having a common chronology for ice cores using known events such as volcanic and solar activity to synchronise the different records.

We agree that it is useful to be able to readily translate between the LD age scale and other scales such as the AICC2012 and WD2014. We have added a table in an appendix to facilitate this translation. This will assist users seeking to use the Law Dome record as part of larger compilations where a common age scale is needed. We have added more details in an additional paragraph now at the end of the section about dating and age horizon uncertainties in the new version of the manuscript.

We do however believe that there is much value in maintaining the Law Dome age scale independently of these. Law Dome is unusual among ice core sites of having such a long record where seasonal variations are able to be clearly discerned in the data.

To weaken the language that the DSS layer-counting errors are conservative.

We accept that the term "conservative" is subjective, and have edited this section to give more detail about the sampling and why we consider the DSS error estimate to be appropriate. DSS is unique as a long record from a high-accumulation East Antarctic site (0.68 m ice equivalent/yr). The mean annual sample resolution at -11 CE is 7 for chemistry, and 19 for oxygen isotopes; sufficient for good quality annual layer counting.

To analyze in high-resolution 10Be around 774 and 993 CE for evaluation of the chronology.

We agree that it would be useful to have this analysis performed in the future. However it is not in the scope of this data paper.

To include a table as supporting information providing the ice-core depths for the previously identified common age markers in NGRIP1, WDC and DSS allowing age-transfer between ice cores.

We agree and now provide such a table in supplementary material which will allow the translation of the DSS age scale to other commonly used ice core age scales as has been suggested here.

Provide some information in the main text that DSS is independent but not synchronous with the WD2014 chronology commonly used for synchronizing ice cores in Antarctica –

and a backbone for volcanic forcing reconstructions. For ongoing initiatives such as PAGES CLIVASH2k this information will be helpful, since it can help to avoid the smoothing and amplitude loss inherent in the stacking of proxy records with limited age synchronization (e.g. PAGES 2k Consortium 2019) which remains a major limitation for our understanding of natural climate variations.

We have added a paragraph in manuscript within the dating and age horizons section that points out that the DSS dating is independent but not synchronous with other commonly used chronologies and reference the age scale translation table that has been added as described above. In addition, we also added a list in the summary section considerations that future users of the data should take into account, we will list this offset in the age scales of the LD data and other Antarctic ice core chronologies in here too.

---

## Editor Decision (ED1)

Very happy to see ice core data products shared via ESSD, thank you.

Access remains unclear. Authors said "We can add instructions to assist users of the data" but have not done so. We need these very clear instructions (if short, at end of abstract as well as in data availability section) on how to bypass registration (email) requirements in order to move forward.

Authors have three options. Doing nothing not an option. Barrier-free access desired.

1) Definitely not true that AADC can not change policy. To my direct knowledge, they have changed policies at least four times since 2006, three times since ESSD started. We do not want to burden authors with changes to institutional policies, but neither can we accept glib author contention that institutions won't change. Change AADC policy impractical, but not impossible.
2) Put the data at Zenodo. A mirror site will help backup policies, Zenodo remains quick and free, many other ESSD data providers use Zenodo, authors could send a clear message to AADC by using Zenodo.
3) Authors can publish clear instructions about how to use the registration-free back door. Bypassing the registration step renders the registration step moot, of course, but authors have little choice. If authors chose this option, they need to provide prominent clearer than clear instructions.

So long as authors will need to make data access changes, they might as well address other deficiencies. Copernicus staff apply excellent typesetting and proofreading skills but authors should save those staff substantial work by making corrections now.

Line 3 "covering over the data perod"? Please correct.

Line 6 "region where few records exist, especially at high temporal resolution" Tense wrong, text awkward, please correct.

Lines 11, 12 "exceptionally high temporal resolution" comes from precip (snow) accumulation rates (mentioned later in this paragraph) not from geographic position. Please correct.

Line 21 Rather than "in particular" I think you mean 'by extension'. Please correct.

Line 23 Why does proximity to Casey station provide benefit? Please clarify.

Line 27 "measurements" "has" tense problem, please correct.

Line 32 annually dated should read as annually-dated, please correct.

Line 32 and following: term "proxies used twice in this sentence, please correct.

Line 37 this sentence is only partially true, depends on whether one addresses ice, ocean, precip, etc. Jones et al. 2016 a very weak and sadly dated (almost obsolete?) reference. Much better list starting from line 40. Needs substantial correction and re-writing.

Line 45 this sentence represents the prime motivation and best justification for the entire effort. Move it forward to a more prominent position?

Line 53 "two levels" of what? Quality control? Temporal resolution? Reader needs to read subsequent sentences? Please change.

Line 54 "quality controlled" should be 'quality-controlled'. Please correct.

Line 55 punctuation error, use sentence break rather than comma. Please correct.

Lines 57-58 confusing run-on sentence. Please correct.

At this point this reader has gone through only 60 of nearly 280 lines of descriptive text, with result that I have found 13 deficiencies / errors. No editor wants to take on such an extensive corrective workload. Please ensure careful reading and multiple corrections of next version by several co-authors, including by at least one native English speaker. We will drive Copernicus proofreaders crazy if we send manuscript forward in such a dismal state. I will read carefully once again, hopefully without stumbling over so many deficiencies.

---

## Author Response (AR2)

**2000 years of annual ice core data from Law Dome, East Antarctica**
**Response to reviewer comments**

We thank the editor for their constructive comments. We have taken time to thoroughly proof-read the manuscript and make some minor corrections. These are best reviewed using the tracked changes version. The larger changes we list below.

*Data Access* The AADC has been in contact with the chief editor on our behalf to clarify if ESSD policy has recently changed and if the AADC policy for access is still acceptable. However, we do agree that the download process may be slightly confusing and barrier-free access desirable, so we have added a sentence in the data availability section to point the reader to the link to "View Dataset Contents".

*Appendix/Supplementary material* While the prevision revision that we uploaded we had removed the appendices from the main text and placed them in supplementary material. This was to ensure that figures and tables did not migrate beyond the end of the main text as had occured in the original version on the manuscript. Those appendices are now included within the one manuscript document at the end. We have also reordered the appendices, putting the translation table between the age scales first.